# SPDIM: Source-Free Unsupervised Conditional and Label Shift Adaptation in EEG

**Shanglin Li**[1,2,†]**, Motoaki Kawanabe**[2,3] **& Reinmar J. Kobler**[2,3,†]
[1] Nara Institute of Science and Technology (NAIST), Nara, Japan
[2] Department of Dynamic Brain Imaging, ATR, Kyoto, Japan
[3] Center for Advanced Intelligence Project, RIKEN, Tokyo, Japan
[†] Equal contribution
`{shanglin,kawanabe,reinmar.kobler}@atr.jp`

## Abstract

The non-stationary nature of electroencephalography (EEG) introduces distribution shifts across domains (e.g., days and subjects), posing a significant challenge to EEG-based neurotechnology generalization. Without labeled calibration data for target domains, the problem is a source-free unsupervised domain adaptation (SFUDA) problem. For scenarios with constant label distribution, Riemannian geometry-aware statistical alignment frameworks on the symmetric positive definite (SPD) manifold are considered state-of-the-art. However, many practical scenarios, including EEG-based sleep staging, exhibit label shifts. Here, we propose a geometric deep learning framework for SFUDA problems under specific distribution shifts, including label shifts. We introduce a novel, realistic generative model and show that prior Riemannian statistical alignment methods on the SPD manifold can compensate for specific marginal and conditional distribution shifts but hurt generalization under label shifts. As a remedy, we propose a parameter-efficient manifold optimization strategy termed SPDIM. SPDIM uses the information maximization principle to learn a single SPD-manifold-constrained parameter per target domain. In simulations, we demonstrate that SPDIM can compensate for the shifts under our generative model. Moreover, using public EEG-based brain-computer interface and sleep staging datasets, we show that SPDIM outperforms prior approaches.

## 1 Introduction

Electroencephalography (EEG) measures multi-channel electric brain activity from the human scalp (Niedermeyer & da Silva, 2005) and can reveal cognitive processes (Pfurtscheller & Da Silva, 1999), emotion states (Suhaimi et al., 2020), and health status (Alotaiby et al., 2014). Neurotechnology and brain-computer interfaces (BCI) aim to extract patterns from the EEG activity that can be utilized for various applications, including rehabilitation and communication (Wolpaw et al., 2002). Despite their capabilities, they currently suffer from a low signal-to-noise ratio (SNR), low specificity, and non-stationarities manifesting as distribution shifts across days and subjects (Fairclough & Lotte, 2020).

For EEG-based neurotechnology, distribution shifts have been traditionally mitigated by collecting labeled calibration data and training domain-specific models (Lotte et al., 2018), limiting neurotechnology utility and scalability (Wei et al., 2022). As an alternative, domain adaptation (DA) learns a model from one or multiple source domains that performs well on different (but related) target domain(s), offering principled statistical learning approaches with theoretical guarantees (Ben-David et al., 2010; Hoffman et al., 2018). Within the BCI field, DA primarily addresses cross-session and cross-subject transfer learning (TL) problems (Wu et al., 2020), aiming to achieve robust generalization across domains (e.g., sessions and subjects) without supervised calibration data.

BCIs that generalize across domains without requiring labeled calibration data are one of the grand challenges in EEG-based BCI research (Fairclough & Lotte, 2020; Wolpaw et al., 2002). Since target domain data is typically unavailable during training, the problem corresponds to a source-free unsu-

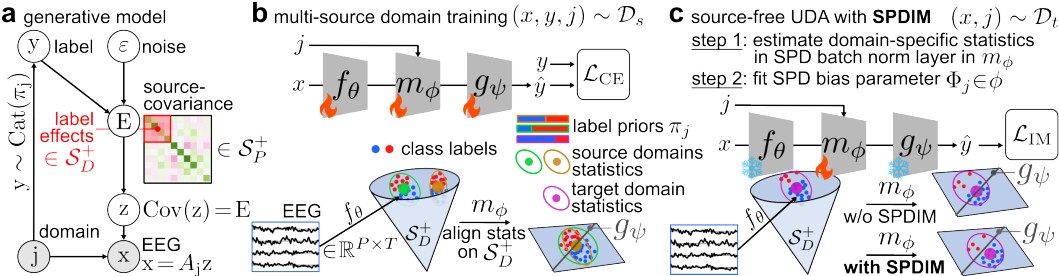

Figure 1: **Framework Overview**. **a**, EEG data x is generated by mixing source signals z with unknown, linear forward models $A_j$. A submanifold $\mathcal{S}_D^+$ of the sources' covariance matrices $E \in \mathcal{S}_P^+$ encodes information about the label $y$. Domain-specific label priors $\pi_j$ and forward models $A_j$ introduce label and conditional distribution shifts, respectively. **b**, Multi-source domain training, utilizes balanced batch sampling and the end-to-end latent alignment framework proposed in (Kobler et al., 2022a). **c**, Proposed SPDIM framework. After latent alignment of marginal distributions (step 1), SPDIM uses the information maximization (IM) loss to fit a bias parameter $\Phi_j \in \mathcal{S}_D^+$ (step 2), and thereby counteract over-corrections in step 1 that are driven by label shifts.

pervised domain adaptation (SFUDA) problem (Liu et al., 2021; Yang et al., 2021). For this problem class, Riemannian geometry-aware statistical alignment frameworks (Barachant et al., 2011) operating with symmetric, positive definite (SPD) matrix-valued features are considered state-of-the-art (SoA) in cross-domain (Roy et al., 2022; Mellot et al., 2023) generalization. They offer several advantageous properties, such as invariance to linear mixing, that are suitable for EEG data (Congedo et al., 2017), as well as consistent (Sabbagh et al., 2020) and inherently interpretable (Kobler et al., 2021) estimators for generative models with a log-linear relationship between the power of latent sources and the labels. Additionally, Collas et al. (2024) suggests that a linear mapping exists between the source and target domains in Riemannian geometry framework. To facilitate generalization across domains, statistical alignment frameworks aim to align first (Zanini et al., 2017; Yair et al., 2019) and second (Rodrigues et al., 2019; Kobler et al., 2022a) order moments, denoted Fréchet mean and variance on Riemannian manifolds. Once the moments are aligned, a model trained on source domains typically generalizes to related target domains (Zanini et al., 2017; Wei et al., 2022; Kobler et al., 2022a; Ju & Guan, 2022; Mellot et al., 2024).

Although infrequently studied, many applications, including EEG-based sleep staging, exhibit label shift (Thölke et al., 2023). Under label shift, aligning the moments of the marginal feature distributions can increase the generalization error (Bakas et al., 2023). To address various sources of distribution shifts in EEG, an SFUDA approach that can deal with additional label shifts is required. Machine learning literature offers several frameworks for SFUDA problems with label shift (Li et al., 2021; Liang et al., 2024), but few have been applied to EEG data. For example, Li et al. (2023) employed the information maximization (Shi & Sha, 2012) objective for cross-domain generalization. Within Riemannian geometry methods, Mellot et al. (2024) studied an EEG-based age regression problem and proposed a framework to facilitate generalization across populations with different prior distributions.

Here, we propose a geometric deep learning framework to tackle SFUDA classification problems under distribution shifts, including label shift. We introduce a realistic generative model with a log-linear relationship between the covariance of latent sources and the labels (Figure 1a). We provide theoretical analyses showing that prior Riemannian statistical alignment methods (Figure 1b) on the SPD manifold can compensate for the conditional distribution shifts introduced in our model but hurt generalization under additional label shifts. As a remedy, we propose a parameter-efficient manifold optimization strategy termed SPDIM. SPDIM employs the information maximization principle to learn a domain-specific SPD-manifold-constrained bias parameter to compensate over-corrections introduced via aligning the Fréchet mean (Figure 1c).

## 2 PRELIMINARIES

### 2.1 SFUDA SCENARIO

Let y and x denote random variables representing the true labels and associated features, and let $p_j(y)$ and $p_j(x|y)$ be the prior and conditional probability distributions for domain $j$. In the transfer learning scenario considered here, we assume that the class priors $\pi_j := p_j(y)$ can be different from each other, as well as specific shifts in the conditional distributions $p_j(x|y)$. We consider a labeled source dataset $\mathcal{D}_s = \{(x_i, y_i, j_i)|x_i \in \mathcal{X}_s, y_i \in \mathcal{Y}_s, j_i \in \mathcal{J}_s\}_{i=1}^{L_s}$ and an unlabeled target dataset $\mathcal{D}_t = \{(x_i, j_i)|x_i \in \mathcal{X}_t, y_i \in \mathcal{Y}_t, j_i \in \mathcal{J}_t\}_{i=1}^{L_t}$, where $y_i$ and $j_i$ (or $j(i)$ if used as suffix) indicate the associated label and domain for each example $i$. We additionally assume that both datasets share the feature space (i.e., $\mathcal{X}_s = \mathcal{X}_t = \mathcal{X}$), contain the same classes (i.e., $\mathcal{Y}_s = \mathcal{Y}_t = \mathcal{Y}$), and comprise examples from different domains (i.e., $\mathcal{J}_s \cap \mathcal{J}_t = \varnothing$). The goal is to transfer the knowledge learned from $\mathcal{D}_s$ to $\mathcal{D}_t$ via first learning a source decoder $h_s$ within hypothesis class $\mathcal{H}$ and then use $h_s$ and the unlabeled target dataset $\mathcal{D}_t$ to learn $h_t \in \mathcal{H}$.

### 2.2 RIEMANNIAN GEOMETRY ON $\mathcal{S}_D^+$

The SPD manifold $\mathcal{S}_D^+ = \{C \in \mathbb{R}^{D \times D} : C^T = C, C \succ 0\}$ together with an inner product on its tangent space $\mathcal{T}_C \mathcal{S}_D^+$ at each point $C \in \mathcal{S}_D^+$ forms a Riemannian manifold. Tangent spaces have Euclidean structure with easy-to-compute distances, which locally approximate Riemannian distances on $\mathcal{S}_D^+$ (Absil et al., 2008). In this work, we consider the affine invariant Riemannian metric (AIRM) $g_C^{\text{AIRM}}(S_1, S_2) = \text{Tr}(C^{-1} S_1 C^{-1} S_2)$ as the inner product, which gives rise to the following distance (Bhatia, 2009):

$$\delta(C_1, C_2) = \| \log(C_1^{-\frac{1}{2}} C_2 C_1^{-\frac{1}{2}}) \|_F \tag{1}$$

where $C_1$ and $C_2$ are two SPD matrices, $\text{Tr}(\cdot)$ denotes the trace, $\log(\cdot)$ the matrix logarithm, and $\| \cdot \|_F$ the Frobenius norm. For a set of points $\mathcal{C} = \{C_i \in \mathcal{S}_D^+\}_{i \leq M}$, the Fréchet mean is defined as the minimizer of the average squared distances:

$$\bar{C} = \arg \min_{G \in \mathcal{S}_D^+} \nu_G(\mathcal{C}) = \arg \min_{G \in \mathcal{S}_D^+} \frac{1}{M} \sum_{i=1}^{M} \delta^2(G, C_i) \tag{2}$$

For $M = 2$, there exists a closed form solution:

$$\bar{C} = \arg \min_{G \in \mathcal{S}_D^+} \nu_G(\{C_1, C_2\}) = C_1 \#_t C_2 |_{t=\frac{1}{2}} = C_1^{\frac{1}{2}} \left( C_1^{-\frac{1}{2}} C_2 C_1^{-\frac{1}{2}} \right)^t C_1^{\frac{1}{2}} \Big|_{t=\frac{1}{2}} \tag{3}$$

where parameter $t \in [0, 1]$ smoothly interpolates along the geodesic $C_1 \#_t C_2$ (i.e., the shortest path) connecting both points.

The logarithmic map $\text{Log}_{\bar{C}} : \mathcal{S}_D^+ \to \mathcal{T}_{\bar{C}} \mathcal{S}_D^+$ and exponential map $\text{Exp}_{\bar{C}} : \mathcal{T}_{\bar{C}} \mathcal{S}_D^+ \to \mathcal{S}_D^+$ project points between the manifold and the tangent space at point $\bar{C}$:

$$\text{Log}_{\bar{C}}(C_i) = \bar{C}^{\frac{1}{2}} \log(\bar{C}^{-\frac{1}{2}} C_i \bar{C}^{-\frac{1}{2}}) \bar{C}^{\frac{1}{2}} \tag{4}$$

$$\text{Exp}_{\bar{C}}(S_i) = \bar{C}^{\frac{1}{2}} \exp(\bar{C}^{-\frac{1}{2}} S_i \bar{C}^{-\frac{1}{2}}) \bar{C}^{\frac{1}{2}} \tag{5}$$

To transport points $S_i \in \mathcal{T}_{\bar{C}} \mathcal{S}_D^+$ from the tangent space at $\bar{C}$ to the tangent space at $\bar{C}_\phi$, parallel transport on $\mathcal{S}_D^+$ can be used as:

$$\Gamma_{\bar{C} \to \bar{C}_\phi}(S_i) = P^\top S_i P, \quad P = \left( \bar{C}^{-1} \bar{C}_\phi \right)^{\frac{1}{2}} \tag{6}$$

While parallel transport is generally defined for tangent space vectors (Absil et al., 2008), for $(\mathcal{S}_D^+, g^{\text{AIRM}})$ it can be directly applied without explicitly computing tangent space projections (i.e., $\text{Exp}_{\bar{C}_\phi} \circ \Gamma_{\bar{C} \to \bar{C}_\phi} \circ \text{Log}_{\bar{C}} = \Gamma_{\bar{C} \to \bar{C}_\phi}$) (Brooks et al., 2019; Yair et al., 2019).

If $\bar{C}_\phi$ lies along the geodesic connecting $\bar{C}$ with the identity matrix $I_D$, there exists a step-size $t \in \mathbb{R}$ so that $\bar{C}_\phi = \bar{C} \#_t I_D$, and (6) simplifies to Mellot et al. (2024):

$$\Gamma_{\bar{C} \to I_D}(C_i; t) = \Gamma_{\bar{C} \to \bar{C} \#_t I_D}(C_i) = \bar{C}^{-\frac{t}{2}} C_i \bar{C}^{-\frac{t}{2}} \tag{7}$$

## 3 METHODS

### 3.1 GENERATIVE MODEL

In the case of EEG, the features $x_i \in \mathcal{X} \subseteq \mathbb{R}^{P \times T}$ comprise epochs of multivariate time-series data with $P$ spatial channels and $T$ consecutive temporal samples. Propagation of brain activity to the EEG electrodes, located at the scalp, is typically modeled as a linear mixture of sources (Nunez & Srinivasan, 2006):

$$x_i := A_{j(i)} z_i \tag{8}$$

where $A_{j(i)} \in \{A \in \mathbb{R}^{P \times P} : \text{rank}(A) = P\}$ is a domain-specific forward model and $z_i \in \mathbb{R}^{P \times T}$ the activity of latent sources. Utilizing the uniqueness of the polar decomposition for invertible matrices, we constrain the model to

$$A_j := Q \exp(P_j) \tag{9}$$

where $Q \in \{Q \in \mathbb{R}^{P \times P} : Q^T Q = I_P\}$ is a orthogonal matrix modeling rotations and $\exp(P_j) \in \mathcal{S}_P^+$ a SPD matrix modeling domain-specific scalings.

Like (Sabbagh et al., 2020; Kobler et al., 2021; Mellot et al., 2024), we consider zero-mean (i.e., $\mathbb{E}\{z_i\} = 0_P \ \forall i$) signals, and a log-linear relationship between the spatial covariance $E_i = \text{Cov}(z_i) \in \mathcal{S}_P^+$ of the latent sources $z_i$ and the target $y_i$. As graphically outlined in Figure 1a, we model the source covariance matrices as:

$$E_i := \exp \circ \text{upper}^{-1}(s_i) \tag{10}$$

where the linear mapping $\text{upper}^{-1} : \mathbb{R}^{P(P+1)/2} \to \mathcal{S}_P$ transforms a vector to a symmetric matrix while preserving its norm (i.e., $S \in \mathcal{S}_P : \|\text{upper}^{-1}(s)\|_F = \|s\|_2$), and latent log-space features $s_i \in R^{P(P+1)/2}$ are generated as:

$$s_i := B\tilde{y}_i + \varepsilon_i, \ \ \tilde{y}_i := 1_{y_i} - \pi_j \tag{11}$$

where $1_{y_i}$ represents one-hot-coded labels, $\pi_j = [p_j(\text{y}=y)]_{y=1,\dots,|\mathcal{Y}|}$ contains the class priors for domain $j$, $\varepsilon_i \in \mathbb{R}^{P(P+1)/2}$ is zero-mean additive noise, and $B \in \{B \in \mathbb{R}^{P(P+1)/2 \times |\mathcal{Y}|} : \text{rank}(B) = |\mathcal{Y}|\}$. We assume that the matrix $B$ is sparse and structured so that label information in $\log(E_i)$ is only encoded in its first $D$-dimensional block (Figure 1a).

**Proposition 1** *Given the specified generative model and a set of examples $\mathcal{E}_j = \{(E_i, y_i, j) | E_i \in \mathcal{S}_P^+, y_i \in \mathcal{Y}\}_{i \leq M_j}$ of domain $j \in \mathcal{J}$, we have that the Fréchet mean of $\mathcal{E}_j$, defined in (2), converges to the identity matrix $I_P$ with $M_j \to \infty$ for all domains $j \in \mathcal{J}$.*

Our proof, provided in appendix A.1, relies on the uniqueness of the Fréchet mean for $(\mathcal{S}_P^+, g^{\text{AIRM}})$ and that $\tilde{y}_i$ and $\varepsilon_i$ in (11) are zero-mean.

For the generated multi-variate time-series features $x_i \in \mathcal{X}$ the empirical covariance matrix $C_i$ is:

$$C_i := \text{Cov}(x_i) = \frac{1}{T} x_i x_i^T \in \mathcal{S}_P^+ \tag{12}$$

Due to the linear relationship between observed features and latent sources (8), we obtain a direct relationship to the latent source covariance matrices $E_i$:

$$C_i = A_{j(i)} E_i A_{j(i)}^T \tag{13}$$

Since $E_i$ encodes the target $y_i \in \mathcal{Y}$ and $A_j$ is invertible, $C_i$ are sufficient descriptors to decode $y_i$.

**Remark 1** *Note that for each domain $j$ the covariance matrices of the generated data $\{C_i : j_i = j\}$ are not necessarily jointly diagonalizable. Depending on the structure of $B$ and $\varepsilon_i$, the proposed generative model reduces to a jointly diagonalizable model if all the off-diagonal elements in $\log(E_i)$ are zero $\forall i$.*

### 3.2 DECODING FRAMEWORK

Given source domain data $\mathcal{D}_s$ and a hypothesis class $\mathcal{H}$, we aim to learn a decoder function $h_s \in \mathcal{H}$, and - once the unlabeled target data $\mathcal{D}_t$ is revealed - use $\mathcal{D}_t$ and $h_s$ to learn $h_t \in \mathcal{H}$. Following

Kobler et al. (2022a), we constrain the hypothesis class $\mathcal{H}$ to functions $h : \mathcal{X} \times \mathcal{J} \rightarrow \mathcal{Y}$ that can be decomposed into a composition of a shared feature extractor $f_\theta : \mathcal{X} \rightarrow \mathcal{S}_D^+$, latent alignment $m_\phi : \mathcal{S}_D^+ \times \mathcal{J} \rightarrow \mathbb{R}^{D(D+1)/2}$, and a shared linear classifier $g_\psi : \mathbb{R}^{D(D+1)/2} \rightarrow \mathcal{Y}$ with parameters $\Theta = \{\theta, \phi, \psi\}$ (Figure 1b). Within this section, we focus on theoretical considerations for $m_\phi$ under our generative model.

**Tangent space mapping (TSM) to recover** $\log(E_i)$   Considering a set of labeled data $\mathcal{D} = \{(x_i, y_i) : x_i \in \mathcal{X}, y_i \in \mathcal{Y}, j_i = j \ \forall i\}$ obtained from a single domain $j$, TSM (Barachant et al., 2011) provides an established (Lotte et al., 2018; Jayaram & Barachant, 2018) decoding approach to infer $y_i$. TSM requires SPD-matrix valued representations. For the considered generative model, covariance features $C_i$, as defined in (12), are a natural choice (i.e., $f_\theta = \text{Cov}$). In a nutshell, TSM first estimates the Fréchet mean $\bar{C}$ of $\mathcal{C} = \{\text{Cov}(x_i) : (x_i, y_i) \in \mathcal{D}\}$, projects each $C_i$ to the tangent space at $\bar{C}$, and finally transports the data to vary around $I_P$. Formally,

$$\tilde{m}_\phi(C_i) := \text{upper} \circ \Gamma_{\bar{C} \rightarrow I} \circ \text{Log}_{\bar{C}}(C_i) = \text{upper}\left(\log\left(\bar{C}^{-\frac{1}{2}} C_i \bar{C}^{-\frac{1}{2}}\right)\right), \ \phi = \{\bar{C}\} \qquad (14)$$

where the resulting representations of $\tilde{m}_\phi$ have a linear relationship to $\log(E_i)$ (Sabbagh et al., 2020; Kobler et al., 2021) and through (10) and (11) also to the labels $y_i$.

**RCT+TSM compensates marginal and conditional shifts**   In the context of functional neuroimaging data, the recentering (RCT) transform (Zanini et al., 2017; Yair et al., 2019) and its extensions (Rodrigues et al., 2019; He & Wu, 2019; Kobler et al., 2022a; Mellot et al., 2023) address source-free UDA problems. Combined with TSM, RCT+TSM essentially applies (14) independently to each domain $j$. The outputs $\tilde{m}_{\phi(j(i))}(C_i)$, where $\phi(j) = \{\bar{C}_j\}$, are treated as domain-invariant and passed on to the shared classifier $g_\psi$.

Although RCT+TSM is an established method to address SFUDA problems for neuroimaging data (Lotte et al., 2018; Wei et al., 2022; Roy et al., 2022), there is a lack of understanding of what kind of distribution shifts RCT+TSM can compensate.

**Proposition 2** *For the generative model, specified in section 3.1, RCT+TSM compensates conditional distribution shifts introduced by the invertible linear map $A_j$, defined in (8), and recovers domain-invariant representations if there are no label shifts (i.e., $p_j(\text{y}=y)=p(\text{y}=y) \ \forall y \in \mathcal{Y}, \ \forall j \in \mathcal{J}_s \cup \mathcal{J}_t$).*

A detailed proof is provided in appendix A.2. Starting with $\tilde{m}_{\phi(j(i))}(C_i)$, as defined in (14) and utilizing the unique polar decomposition (9) of $A_j$ into rotational $Q$ and scaling $\exp(P_j)$ transformations along with proposition 1 we obtain:

$$\log\left(\bar{C}_{j(i)}^{-\frac{1}{2}} C_i \bar{C}_{j(i)}^{-\frac{1}{2}}\right) = Q \log(E_i) Q^T = Q \text{upper}^{-1}\left(B\left(1_{y_i} - \pi_{j(i)}\right) + \varepsilon_i\right) Q^T \qquad (15)$$

If there are no label shifts we have $\pi_j = \pi_k \ \forall j, k \in \mathcal{J}_s \cup \mathcal{J}_t$. Then (15) only contains domain-invariant terms on the right hand side. Thus, RCT+TSM compensates the conditional shifts introduced by $\exp(P_j)$. $\square$

**Remark 2** *If all sources in (8) can be partitioned into relevant and irrelevant sources $z_i = [z_i^{\|}, z_i^{\perp}], z_i^{\|} = f(y_i) \in \mathbb{R}^{D \times T}, z_i^{\perp} \neq f(y_i) \in \mathbb{R}^{(P-D) \times T}$ that are independent from each other, then the source covariance matrices $E_i$ have a block-diagonal structure. Consequently, RCT+TSM compensates marginal shifts in $z_i^{\perp}$ and conditional shifts in $z_i^{\|}$ introduced by $\exp(P_j)$.*

**Alignment under label shifts**   We aim to extend RCT+TSM to extract label shift invariant representations. Specifically, we aim to apply additional transformations on $(\mathcal{S}_D^+, g.^{\text{AIRM}})$ that attenuate the effect of class priors $\pi_j$ in (15). We first rewrite (15) as:

$$\bar{C}_{j(i)}^{-\frac{1}{2}} C_i \bar{C}_{j(i)}^{-\frac{1}{2}} = \exp\left(Q \log(E_i) Q^T\right) = \exp\left(Q \text{upper}^{-1}\left(B\left(1_{y_i} - \pi_{j(i)}\right) + \varepsilon_i\right) Q^T\right) \qquad (16)$$

$$= \exp\left(Q \underbrace{\text{upper}^{-1}\left(B 1_{y_i} + \varepsilon_i\right)}_{\log(\tilde{E}_i)} Q^T - Q \underbrace{\text{upper}^{-1}\left(B \pi_{j(i)}\right)}_{\bar{P}_{j(i)}} Q^T\right) \qquad (17)$$

where we split $\log(E_i)$ into domain-invariant $\log(\tilde{E}_i)$ and label shift $\bar{P}_{j(i)}$ terms. To separate both terms into products of matrices, we utilize $\exp(\alpha(A+B)) = \exp(\alpha B/2)\exp(\alpha A)\exp(\alpha B/2) + \mathcal{O}(\alpha^3)$ (Higham, 2008, Theorem 10.5), resulting in:

$$\bar{C}_{j(i)}^{-\frac{1}{2}} C_i \bar{C}_{j(i)}^{-\frac{1}{2}} = Q \exp\left(\bar{P}_{j(i)}\right)^{-\frac{1}{2}} Q^T Q \tilde{E}_i Q^T Q \exp\left(\bar{P}_{j(i)}\right)^{-\frac{1}{2}} Q^T + \mathcal{O}(\|\log(E_i)\|_F^3) \quad (18)$$

with the approximation error $\mathcal{O}(\|\log(E_i)\|_F^3)$ decaying cubically for $\|\log(E_i)\|_F < 1$. In this form, it is straightforward to see that an additional bilinear transformation on the left hand side in (18) with an SPD matrix can approximately compensate the effect of $\exp(\bar{P}_{j(i)})$. We denote this parameter as domain-specific bias parameter $\Phi_{j(i)} \in \mathcal{S}_D^+$, and generalize RCT+TSM to:

$$m_{\phi(j(i))}(C_i) = \text{upper} \circ \log\left(\Phi_{j(i)}^{\frac{1}{2}} \bar{C}_{j(i)}^{-\frac{1}{2}} C_i \bar{C}_{j(i)}^{-\frac{1}{2}} \Phi_{j(i)}^{\frac{1}{2}}\right), \ \phi(j(i)) = \{\bar{C}_{j(i)}, \Phi_{j(i)}\} \quad (19)$$

Note that if $\exp(\bar{P}_j)$ and $\exp(P_j)$ share the same eigenvectors, they commute and lie on the same geodesic connecting $\exp(P_j)$ with $I_D$. Consequently, the combined effect of $\exp(\bar{P}_j)$ and $\exp(P_j)$ is constrained to the geodesic connecting $\bar{C}_j$ with $I_D$. Then, the solution space can be constrained to $\Phi_j \in \{\Phi : \Phi \in \mathcal{S}_D^+, \Phi = \bar{C}_j \#_{\varphi_j} I_D, \varphi_j \in \mathbb{R}\}$, and (7) used to simplify (19) to:

$$m_{\phi(j(i))}^{\#}(C_i) = \text{upper} \circ \log \circ \Gamma_{\bar{C}_{j(i)} \to \bar{C}_{j(i)} \#_{\varphi_{j(i)}} I_D}(C_i), \ \phi(j(i)) = \{\bar{C}_{j(i)}, \varphi_{j(i)}\} \quad (20)$$

where the geodesic step-size parameter $\varphi_{j(i)} \in \mathbb{R}$ needs to be learned.

**Latent alignment with domain-specific SPD batch norm.** Parametrizing the feature extractor $f_\theta : \mathcal{X} \to \mathcal{S}_D^+$ as a neural network naturally extends the decoding framework to neural networks with SPD matrix-valued features (Huang & Gool, 2017). In this end-to-end learning setting, SPD batch norm (SPDBN)(Brooks et al., 2019; Kobler et al., 2022b) and domain-specific batch norm (Kobler et al., 2022a) layers can be utilized to implement $m_\phi$.

## 3.3 SPD MANIFOLD INFORMATION MAXIMIZATION

We utilize the labeled source domain dataset $\mathcal{D}_s$ to learn the shared feature extractor $f_\theta$, $g_\psi$, and $m_\phi$ for the source domains $j \in \mathcal{J}_s$ with the cross-entropy loss as the training objective (Figure 1b).

For the target domains, we keep $f_\theta$ and $g_\psi$ fixed and learn $\phi(j) \forall j \in \mathcal{J}_t$ (Figure 1c). For each domain $j \in \mathcal{J}_t$, we use (2) to compute the Fréchet mean $\bar{C}_{j(i)}$ of $\mathcal{C}_j = \{f_\theta(x_i) : (x_i, j(i)) \in \mathcal{D}_t, j(i) = j\}$. To estimate $\Phi_{j(i)}$ in an unsupervised fashion, we employ the information maximization (IM) loss (Shi & Sha, 2012) to ensure that target outputs are individually certain and globally diverse. The IM loss is a popular training objective for SFUDA and test-time adaptation frameworks (Liang et al., 2020; 2024). In practice, when the target domain data is revealed, we initialize $\Phi_{j(i)}$ with $I_D$ and $\varphi_{j(i)}$ with 1. We then minimize the following $\mathcal{L}_{\text{CEM}}$ and $\mathcal{L}_{\text{MEM}}$ that together constitute the $\mathcal{L}_{\text{IM}}$ loss:

$$\mathcal{L}_{\text{IM}} = \mathcal{L}_{\text{CEM}} + \mathcal{L}_{\text{MEM}} \quad (21)$$

$$\mathcal{L}_{\text{CEM}} = -\mathbb{E}_{(x_i, j_i) \in \mathcal{D}_t} \left\{ \sum_{k=1}^{|\mathcal{Y}|} \delta_k\left(\frac{h_t(x_i, j_i)}{T}\right) \log \delta_k\left(\frac{h_t(x_i, j_i)}{T}\right) \right\}$$

$$\mathcal{L}_{\text{MEM}} = \sum_{k=1}^{|\mathcal{Y}|} \hat{p}_k \log \hat{p}_k, \ \hat{p}_k = \mathbb{E}_{(x_i, j_i) \in \mathcal{D}_t} \left\{ \delta_k\left(\frac{h_t(x_i, j_i)}{T}\right) \right\}$$

where $\delta_k$ is the k-th element of the softmax output, $\mathcal{L}_{\text{CEM}}$ is conditional entropy minimization, $\mathcal{L}_{\text{MEM}}$ is marginal entropy maximization and factor $T$ is temperature scaling. IM balance is more effective than conditional entropy minimization because minimizing only the conditional entropy may lead to model collapse with all test data allocated to one class (Grandvalet & Bengio, 2004). We additionally employed a temperature scaling factor $T$ to adjust the model's prediction confidence on the target data, a common technique for calibrating probabilistic models (Li et al., 2023; Guo et al., 2017). Temperature scaling uses a single scalar parameter $T > 0$ for all classes, and it increases the softmax output entropy when $T > 1$ and decreases it when $0 < T < 1$.

## 4 EXPERIMENTS

We conducted simulations and experiments with public EEG motor imagery and sleep stage datasets to evaluate our proposed framework empirically.

**Multi-source domain training** Following Kobler et al. (2022a), we parameterize $h_s = g_\psi \circ m_\phi \circ f_\theta$ as a neural network and fit the source decoder $h_s$ in an end-to-end fashion (Figure 1b), denoted as TSMNet. We used the standard-cross entropy loss as training objective, and optimized the parameters with the Riemannian ADAM optimizer (Bécigneul & Ganea, 2018). We split the source domains' data into training and validation sets (80% / 20% splits, randomized, stratified by domain and label) and iterated through the training set for 100 epochs. We stick to the TSMNet hyper-parameters as provided in the public reference implementation (for implementation details see Appendix A.7.1).

**SPDIM Source-free domain adaptation** SPDIM keeps the fitted source feature extractor $f_\theta$ and linear classifier $g_\psi$ fixed and estimates a domain-specific bias parameter for latent alignment $m_\phi$ (Figure 1c). Depending on the choice of the bias parameter, we distinguish between SPDIM(bias), defined in (19), and SPDIM(geodesic), defined in (20). We use the entire target domain data to estimate gradients for the IM loss (21) and Riemannian ADAM to optimize the bias parameter for 50 epochs.

**SFUDA Baseline Methods.** We consider several multi-source (-target) SFUDA baseline methods, including Recenter (RCT) (Zanini et al., 2017), Euclidean alignment (EA) (He & Wu, 2019), and spatio-temporal Monge alignment (STMA) (Gnassounou et al., 2024). These alignment methods are model-agnostic techniques that are applied to the EEG data before a classifier is fitted. Among the end-to-end learning SFUDA methods, we consider SPDDSBN (Kobler et al., 2022a) which was introduced together with the TSMNet architecture. Lastly, we compare SPDIM to classic IM approaches (Shi & Sha, 2012) that adapt parameters in $f_\theta$ or $g_\psi$.

**No DA Baseline Methods.** Methods in this category, denoted w/o SFUDA, treat the problem as a standard supervised learning problem; they do not utilize the domain labels $\mathcal{J}_s \cup \mathcal{J}_t$ during training and testing. In models that perform TSM, all data are projected to the tangent space at the Fréchet mean of the entire source dataset $\mathcal{D}_s$.

We used publicly available Python code for baseline methods and implemented custom methods using the packages torch (Paszke et al., 2019), scikit-learn (Pedregosa et al., 2011), braindecode (Schirrmeister et al., 2017), geoopt (Kochurov et al., 2020) and pyRiemann Barachant et al. (2023). We conducted the experiments on standard computation PCs with 32-core CPUs, 128 GB of RAM, and a single GPU.

### 4.1 SIMULATIONS

To examine the effectiveness of SPDIM, we simulated binary classification problems under our generative model (implementation details in Appendix A.6). We used balanced accuracy as evaluation metric and examined the performance of SPDIM(bias) and SPDIM(geodesic) against RCT (Zanini et al., 2017) over different label ratios. Figure 2 summarizes the results for different class separability levels. Supplementary Figures display the methods' performance over parameters $|\mathcal{J}_s|$ (Figure A1), $M_j$ (Figure A2), $P$ (Figure A3), and $D$ (Figure A4).

The simulation results empirically confirm Proposition 2. That is, RCT can compensate the conditional shifts introduced by $A_j$ if there are no labels shifts (i.e., label ratio = 1.0). They also demonstrate that as the label shifts become more severe (i.e., lower label ratio), the average performance of SPDIM decreases while the variability increases. Still, SPDIM outperforms RCT across almost all considered parameter configurations.

### 4.2 EEG MOTOR IMAGERY DATA

Almost all public motor imagery datasets are generated in a highly controlled lab environment and desgined to be balanced. However, in realistic brain-computer interface application settings, the variability of human behavior and environmental factors likely cause label shifts across days and subjects. To bridge the gap between controlled research settings and real-world scenarios, we artificially introduced label shifts in the target domains.

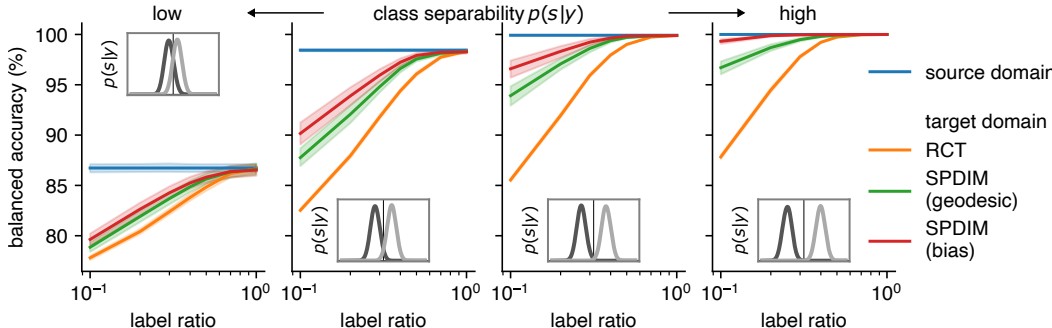

Figure 2: **Simulation results.** Balanced accuracy scores (higher is better) across target domain label ratios (i.e., majority to minority label ratio) on the x-axis and class separability $p(s|y)$ across panels. Source domain labels were balanced.

We considered 4 public motor imagery datasets: BNCI2014001 (Tangermann et al., 2012) (9 subjects/2 sessions/4 classes/22 channels), BNCI2015001 (Faller et al., 2012) (12/2-3/2/13), Zhou2016 (Zhou et al., 2016) (4/3/3/14), and BNCI2014004 (Leeb et al., 2007) (9/5/2/3). We used MOABB (Jayaram & Barachant, 2018; Chevallier et al., 2024) to pre-process the continuous time-series data and extract labeled epochs. Pre-processing included resampling EEG signals to 250 or 256 Hz, applying temporal filters to capture frequencies between 4 and 36 Hz, and extracting 3-second epochs linked to specific class labels. Following Kobler et al. (2022a), we use TSMNet as model architecture and treat sessions as domains, and use a leave-one-group-out cross-validation (CV) scheme to fit and evaluate the methods. To evaluate cross-session transfer, we fitted and evaluated models independently per subject and treated the session as the grouping variable. To evaluate cross-subject transfer, we treated the subject as the grouping variable. After running pilot experiments with the BNCI2014001 dataset, we set the temperature scaling factor in (21) to $T = 2$ for binary classification problems and $T = 0.8$ otherwise. Early stopping were fit with a single stratified (domain and labels) inner train/validation split. We used balanced accuracy as the metric to examine the performance of each method at label ratios of 1.0 and 0.2 for the source and target domains, respectively.

Grand average results across all 4 datasets are summarized in Figure 3, and grouped by the transfer learning scenario (cross-session, cross-subject). To attenuate large variability across subjects, we report scores relative to the score obtained with SPDDSBN w/o label shifts (i.e., 1.0 label ratio). Detailed results per dataset are listed in Table A4 (w/ label shifts) and Table A4 (w/o label shifts) along with the ones of relevant baseline methods. Although no method can perfectly compensate the artificially introduced label shifts, SPDIM(bias) is consistently at the top (cross-subject) or among the top (cross-session) performing methods. Significance testing (n=34 subjects), summarized in Table A4, revealed that the performance of SPDIM(bias) is significantly higher than w/o, SPDDSBN, IM(classifier), and IM(all) in the cross-subject setting as well as SPDDSBN, IM(classifier), and IM(all) in the cross-session scenario.

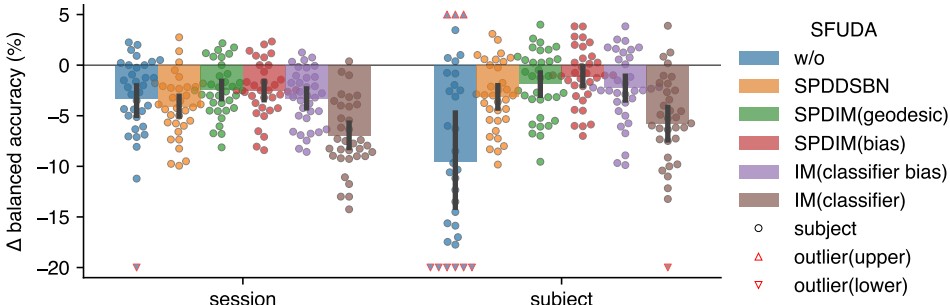

Figure 3: **Motor-imagery results (0.2 label ratio).** Average of test-set scores (balanced accuracy; higher is better; error bars indicate 95% confidence interval) relative to TSMNet+SPDSBN (Kobler et al., 2022a) w/o label shifts. For extended results per dataset, see Tables A4 and A5.

### 4.3 EEG-BASED SLEEP STAGING

The aim of this experiment is to demonstrate the effectiveness of our method on datasets with inherent label shifts. Due to the inherent variability of sleep, most sleep stage datasets exhibit label shifts across subjects (Eldele et al., 2023). Sleep stage classification plays a key role in assessing sleep quality and diagnosing sleep disorders (Perez-Pozuelo et al., 2020). Yet, automated frameworks lack accuracy and suffer from poor generalization across domains, resulting in accuracy drops compared to expert neurologists.

We considered 4 public sleep stage datasets: CAP (Terzano et al., 2001; Goldberger et al., 2000), Dreem (Guillot et al., 2020), HMC (Alvarez-Estevez & Rijsman, 2021a;b), and ISRUC (Khalighi et al., 2016). A detailed description is provided in Supplementary Table A1. We consider sleep stages following the AASM (Berry et al., 2012) standard (W, N1, N2, N3, REM). If data was originally scored following the K&M (Wolpert, 1969) standard (W, S1, S2, S3, S4, REM), we merged stages S3 and S4 into a single stage N3. EEG data pre-processing followed (Guillot & Thorey, 2021) and was implemented with MNE-python (Gramfort et al., 2014). First, all EEG channels were retained, and an IIR band-pass filter ranging from 0.2 to 30 Hz was applied. The signals were then resampled to 100 Hz, and non-overlapping 30-second epochs were extracted together with the associated labels. To attenuate the effects of gross outliers, each recording was scaled to have unit inter-quartile range and a median of zero, and values exceeding 20 times the inter-quartile range were clipped Perslev et al. (2021). Lastly, subjects with corrupted data (e.g., mismatched labels and epochs) were excluded, resulting in a total of 426 remaining subjects.

Since label shifts occur in the source domains of sleep stage data, the considered models are trained with a balanced mini-batch sampler, which is a popular method to compensate for label shifts during training in deep learning (Cao et al., 2019). The balanced sampler over-sampled minority classes to ensure that the label distribution per domain is balanced within each mini-batch.

To evaluate the methods, we employed a 10-fold grouped cross-validation scheme, ensuring that each group (i.e., subject) appears either in the training set (i.e., source domains) or the test set (i.e., target domains). As before, we set the temperature scaling factor $T = 0.8$ and used stratified (labels and domains) inner train/validation splits for early stopping. In addition to the TSMNet architecture, we included four baseline deep learning architectures specifically proposed for sleep staging: Chambon (Chambon et al., 2018), Usleep (Perslev et al., 2021), DeepSleepNet (Supratak et al., 2017), and AttnNet (Eldele et al., 2023) (implementation details in Appendix A.7.2).

Table 1 summarizes the results across datasets along with the grand average results of published baseline methods. Extended results for all considered baseline methods are listed in Supplementary Table A2. TSMNet+SPDIM(bias) significantly outperforms all other methods for the patient and healthy (except TSMNet+STMA) subject groups. Overall, the margin to TSMNet+SPDDSBN was approx. 5% in the patient group, which indicates that SPDIM has great potential for clinical applications.

**Ablation Study** Table 2 summarizes grand average test scores relative to SPDIM(bias). We highlight four observations. First, all considered ablations lead to a significant performance drop of at least 3% compared to SPDIM(bias), suggesting the combined importance of IM paired with the manifold-constrained bias parameter. Second, in the presence of label shifts, fitting $\tilde{m}_\phi$ per domain (i.e., TSMNet+SPDSBN) hurts generalization compared to global $\tilde{m}_\phi$ (i.e., TSMNet+w/o). Third, fine-tuning the classifier bias parameter yields approximately the same performance as SPDIM(geodesic), extending the finding of (Mellot et al., 2024) from regression to classification scenarios. Fourth, IM methods obtained the top 3 scores, but two variants failed to improve upon the global method, which underscores the importance of regularization (i.e., via selecting the right parameter) to prevent the IM loss from overfitting.

## 5 DISCUSSION

We proposed a geometric deep learning framework, denoted SPDIM, to address SFUDA problems with conditional and label shifts and demonstrated its utility in highly relevant EEG-based neurotechnology application scenarios. We first introduced a realistic generative model, and provided theoretical analyses showing that prior Riemannian statistical alignment methods that align

| Model | Group: SFUDA | CAP patient (n=82) | Dreem healthy (n=22) | Dreem patient (n=50) | HMC patient (n=154) | ISRUC healthy (n=10) | ISRUC patient (n=108) | Overall healthy (n=32) | Overall patient (n=394) |
|---|---|---|---|---|---|---|---|---|---|
| Chambon | w/o | • 64.7 (10.7) | • 65.2 (9.1) | • 53.4 (16.2) | • 64.0 (9.6) | 72.7 (3.9) | • 68.9 (8.2) | • 67.5 (8.6) | • 64.2 (11.5) |
|  | EA | • 63.8 (11.3) | • 64.7 (10.3) | • 54.5 (14.4) | • 63.4 (10.7) | 74.9 (4.8) | 70.7 (9.3) | • 67.9 (10.0) | • 64.3 (12.0) |
|  | STMA | • 65.2 (10.0) | • 67.1 (7.7) | • 51.3 (19.0) | • 63.8 (9.1) | 74.8 (4.4) | 70.8 (7.6) | • 69.5 (7.7) | • 64.4 (12.1) |
| USleep | w/o | • 59.1 (10.3) | • 55.9 (8.3) | • 48.4 (8.2) | · 66.6 (9.3) | 72.9 (6.1) | • 68.8 (8.3) | • 61.3 (11.0) | • 63.3 (11.3) |
| Deep-SleepNet | w/o | · 68.1 (11.3) | · 68.6 (9.9) | **68.7** (10.3) | • 64.9 (10.3) | 75.8 (4.5) | 72.7 (9.1) | 70.9 (9.1) | • 68.2 (10.7) |
| AttnSleep | w/o | 68.9 (10.3) | • 65.4 (10.7) | · 61.7 (14.2) | 67.7 (10.0) | 75.4 (4.0) | **73.1** (8.6) | • 68.5 (10.2) | 68.7 (10.9) |
| TSMNet | w/o | • 68.0 (11.2) | • 65.9 (9.6) | 66.7 (12.4) | • 63.6 (11.3) | · 73.4 (3.9) | • 68.7 (9.3) | • 68.3 (8.9) | • 66.3 (11.1) |
|  | SPDDSBN | • 68.2 (8.8) | • 68.9 (5.8) | • 64.2 (8.5) | • 62.2 (6.0) | · 72.8 (2.8) | • 66.6 (6.4) | • 70.1 (5.3) | • 64.9 (7.5) |
|  | SPDIM(bias) *(proposed)* | **71.0** (9.6) | **72.1** (8.0) | 68.1 (9.8) | **68.6** (8.5) | **76.7** (3.4) | 71.6 (6.9) | **73.5** (7.2) | **69.9** (8.6) |

Table 1: **Sleep-staging results per dataset.** Average of test-set scores (balanced accuracy; higher is better; standard-deviation in brackets). Permutation-paired t-tests were used to identify significant differences between TSMNet+SPDIM (*proposed*) and baseline methods (1e4 permutations, 14 tests, t-max correction). Significant differences are highlighted (· $p \leq 0.05$, • $p \leq 0.01$, • $p \leq 0.001$). Extended results are provided in Table A2.

| Alignment | fit Fréchet mean | $\mathcal{L}_{IM}$ | $\Theta_{IM}$ | mean (std) | t-val (p-val) |
|---|---|---|---|---|---|
| $m_\phi$ (19) | per domain $j$ | ✓ | $\Phi_j$ | - | - |
| $m_\phi^\#$ (20) | per domain $j$ | ✓ | $\varphi_j$ | -3.0 (3.6) | -17.4 (0.0001) |
| $\tilde{m}_\phi$ (14) | per domain $j$ | ✗ | - | -4.8 (4.5) | -22.1 (0.0001) |
| $\tilde{m}_\phi$ (14) | global (i.e., $\mathcal{D}_s$) | ✗ | - | -3.7 (8.5) | -9.0 (0.0001) |
| $\tilde{m}_\phi$ (14) | per domain $j$ | ✓ | bias in $\psi$ | -3.0 (3.8) | -16.1 (0.0001) |
| $\tilde{m}_\phi$ (14) | per domain $j$ | ✓ | $\psi$ | -4.8 (5.4) | -18.5 (0.0001) |
| $\tilde{m}_\phi$ (14) | per domain $j$ | ✓ | $\theta \cup \psi$ | -27.1 (13.5) | -41.5 (0.0001) |

Table 2: **Sleep-staging ablation results.** Balanced accuracy scores (higher is better) relative to the proposed method. Averages and standard deviation summarize the individual (n=426 subjects) test-set scores. Student's t values and adjusted p values indicate the effect strength (permutation-paired t-tests, 1e4 permutations, 6 tests, t-max correction). Table A3 lists results per dataset and group.

the Fréchet mean can compensate for the conditional distribution shifts introduced in our generative model, but hurt generalization under additional label shifts. As a remedy, we proposed SPDIM to learn a domain-specific SPD manifold-constrained bias parameter to compensate for over-corrections introduced via aligning the Fréchet means by employing the information maximization principle. In simulations and experiments with real EEG data, SPDIM consistently achieved the highest scores among the considered baseline methods.

A limitation of our framework is that the IM loss, due to large noise and outliers, can sometimes estimate an inappropriate bias parameter, leading to the data being shifted in the wrong direction. While it generally improves average performance, it also increases variability. We expect future work to explore a more robust way to estimate the bias parameter.

## 6 Acknowledgments

Motoaki Kawanabe and Reinmar J Kobler were partially supported by the Innovative Science and Technology Initiative for Security Grant Number JPJ004596, ATLA and KAKENHI (Grants-in-Aid for Scientific Research) under Grant Numbers 21K12055, JSPS, Japan.

## 7 Author Contributions

Shanglin Li contributed to the study under the co-supervision of Motoaki Kawanabe and Reinmar J. Kobler. Shanglin Li and Reinmar J. Kobler developed the decoding methods. Reinmar J. Kobler contributed the theoretical analysis. Shangling Li and Reinmar J. Kobler performed the simulation experiments. Shanglin Li conducted the experiments with EEG data. The draft of the manuscript was written by Shanglin Li and Reinmar Kobler. All authors read and approved the final manuscript.

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

# A    APPENDIX

## A.1    PROOF OF PROPOSITION 1

Proposition 1 stats that given the generative model specified in section 3.1 and a set of examples $\mathcal{E}_j = \{(E_i, y_i, j) | E_i \in \mathcal{S}_P^+, y_i \in \mathcal{Y}\}_{i \leq M_j}$ of domain $j \in \mathcal{J}$, we have that the Fréchet mean of $\mathcal{E}_j$, defined in (2), converges with $M_j \to \infty$ to the identity matrix $I_P$ for all domains $j \in \mathcal{J}$.

Since $(\mathcal{S}_P^+, g^{\text{AIRM}})$ forms a Cartan-Hadamard manifold with global non-positive sectional curvature, a unique Fréchet mean exists (Bhatia, 2013). At the global minimum, we have:

$$0 \stackrel{!}{=} \nabla_E \nu_E(\mathcal{E}_j) = \nabla_E \frac{1}{M_j} \sum_{i=1}^{M_j} \delta^2(E, E_i) = \frac{1}{M_j} \sum_{i=1}^{M_j} \nabla_E \delta^2(E, E_i) = -\frac{1}{M_j} \sum_{i=1}^{M_j} \text{Log}_E(E_i) \quad (22)$$

$$= -\frac{1}{M_j} \sum_{i=1}^{M_j} E^{\frac{1}{2}} \log(E^{-\frac{1}{2}} E_i E^{-\frac{1}{2}}) E^{\frac{1}{2}} = -\frac{1}{M_j} E^{\frac{1}{2}} \left( \sum_{i=1}^{M_j} \log(E^{-\frac{1}{2}} E_i E^{-\frac{1}{2}}) \right) E^{\frac{1}{2}} \quad (23)$$

$$0 \stackrel{!}{=} \sum_{i=1}^{M_j} \log(E^{-\frac{1}{2}} E_i E^{-\frac{1}{2}}) \quad (24)$$

where we used the derivative of the Riemannian distance (Moakher, 2005; Pennec, 2018) in (22). For $E = I_P$, (24) simplifies to:

$$\sum_{i=1}^{M_j} \log(E_i) \stackrel{!}{=} 0 \underset{M_j \to \infty}{\Longrightarrow} \mathbb{E}\{\log(E_i)\} \stackrel{!}{=} 0 \quad (25)$$

$$\mathbb{E}\{B\tilde{y}_i + \varepsilon_i\} = B\mathbb{E}\{\tilde{y}_i\} + \mathbb{E}\{\varepsilon_i\} \stackrel{!}{=} 0 \quad (26)$$

which holds true because by definition $\tilde{y}_i$ and $\varepsilon_i$ are zero-mean. Therefore, $E_j = I_P$ is the global minimizer of $\nu_E(\mathcal{E}_j)$ for all domains $j \in \mathcal{J}$. $\square$

## A.2    PROOF OF PROPOSITION 2

Proposition 2 states that for the generative model, specified in section 3.1, RCT+TSM compensates conditional distribution shifts introduced by the invertible linear map $A_j$, defined in (8), and recovers domain-invariant representations if there are no target shifts (i.e., $p_j(\text{y=}y) = p(\text{y=}y) \, \forall y \in \mathcal{Y}, \, \forall j \in \mathcal{J}_s \cup \mathcal{J}_t$).

Due to the congruence invariance of the Fréchet mean (Bhatia, 2013) we have

$$\bar{C}_j = \arg \min_{G \in \mathcal{S}_P^+} \nu_G(\mathcal{C}_j) = A_j \bar{E}_j A_j^T \quad (27)$$

where $\bar{E}_j$ is the Fréchet mean of $\mathcal{E}_j$. Utilizing proposition 1 (i.e., $\bar{E}_j = I_P \, \forall j$) and plugging in (9) for $A_j$, (27) simplifies to:

$$\bar{C}_j = A_j I_P A_j^T = Q \exp(P_j) \exp(P_j) Q^T = Q \exp(P_j)^2 Q^T \quad (28)$$

$$\Rightarrow \bar{C}_j^{-\frac{1}{2}} = Q \exp(P_j)^{-1} Q^T \quad (29)$$

The RCT+TSM transform, as defined in (14), recenters the data $C_i \in \mathcal{C}_j$ for each domain $j$. Excluding the invertible mapping upper, RCT+TSM computes:

$$\log \left( \bar{C}_j^{-\frac{1}{2}} C_i \bar{C}_j^{-\frac{1}{2}} \right) = \log \left( Q \exp(P_j)^{-1} Q^T C_i Q \exp(P_j)^{-1} Q^T \right) \quad (30)$$

$$= \log \left( Q \exp(P_j)^{-1} A_j E_i A_j^T \exp(P_j)^{-1} Q^T \right) \quad (31)$$

$$= \log \left( Q \exp(P_j)^{-1} Q^T Q \exp(P_j) E_i \exp(P_j) Q^T Q \exp(P_j)^{-1} Q^T \right) \quad (32)$$

$$= \log \left( Q E_i Q^T \right) = Q \log(E_i) Q^T \quad (33)$$

where we used (13) for $C_i$, the fact that $Q$ is an orthogonal matrix and $\log(ACA^{-1}) = A\log(C)A^{-1}$ for non-singular $A$ and $C \in \mathcal{S}_P^+$. Plugging (11) for $\log(E_i)$, we obtain a direct relationship to the label $y_i$

$$\log\left(\bar{C}_j^{-\frac{1}{2}} C_i \bar{C}_j^{-\frac{1}{2}}\right) = Q\text{upper}^{-1}\left(B\left(1_{y_i} - \pi_j\right) + \varepsilon_i\right) Q^T \tag{34}$$

If there are no target shifts the class priors are constant $p_j(\mathrm{y}{=}y) = p(\mathrm{y}{=}y) \; \forall y \in \mathcal{Y}$ for all domains $j \in \mathcal{J}_s \cup \mathcal{J}_t$, and consequently $\pi_j = \pi$. Then (34) simplifies to:

$$\log\left(\bar{C}_j^{-\frac{1}{2}} C_i \bar{C}_j^{-\frac{1}{2}}\right) = Q\text{upper}^{-1}\left(B\left(1_{y_i} - \pi\right) + \varepsilon_i\right) Q^T \tag{35}$$

which contains only domain-invariant terms on the right hand side. Thus, RCT+TSM compensates the conditional shifts introduced by $A_j$. $\square$

### A.3   SLEEP STAGE DATASET DETAILS

| Dataset | Recordings | Subjects | Channel numbers | Patients | Scorer | Scoring rule |
|---------|-----------|----------|-----------------|----------|--------|--------------|
| ISRUC-SG1 | 100 | 100 | 6 | ✓ | Scorer1 | AASM |
| ISRUC-SG2 | 16 | 8 | 6 | ✓ | Scorer1 | AASM |
| ISRUC-SG3 | 10 | 10 | 6 | ✗ | Scorer1 | AASM |
| Dreem-SG1 | 22 | 22 | 12 | ✗ | Scorer1 | AASM |
| Dreem-SG2 | 50 | 50 | 8 | ✓ | Scorer1 | AASM |
| HMC | 154 | 154 | 4 | ✓ | - | AASM |
| CAP-SG1 | 36 | 36 | 13 | ✓ | - | K&M |
| CAP-SG2 | 34 | 34 | 9 | ✓ | - | K&M |
| CAP-SG3 | 22 | 22 | 5 | ✓ | - | K&M |

Table A1: **Sleep stage dataset details.** Overview of datasets and their subgroups (SG), including the number of recordings, subjects, channel numbers, patient status, scorer (the chosen scorer if there are multiple scorers), and scoring rule.

## A.4 EEG-based Sleep Staging Results

| Model | SFUDA | | CAP patient (n=82) | Dreem healthy (n=22) | Dreem patient (n=50) | HMC patient (n=154) | ISRUC healthy (n=10) | ISRUC patient (n=108) | Overall healthy (n=32) | Overall patient (n=394) |
|---|---|---|---|---|---|---|---|---|---|---|
| Chambon | w/o | mean | 64.7 | 65.2 | 53.4 | 64.0 | 72.7 | 68.9 | 67.5 | 64.2 |
| | | std | 10.7 | 9.1 | 16.2 | 9.6 | 3.9 | 8.2 | 8.6 | 11.5 |
| | | t-val | ●-6.8 | ●-5.3 | ●-6.1 | ●-7.0 | -2.6 | ●-4.6 | ●-5.8 | ●-11.3 |
| | EA | mean | 63.8 | 64.7 | 54.5 | 63.4 | 74.9 | 70.7 | 67.9 | 64.3 |
| | | std | 11.3 | 10.3 | 14.4 | 10.7 | 4.8 | 9.3 | 10.0 | 12.0 |
| | | t-val | ●-7.5 | ●-3.8 | ●-6.1 | ●-7.5 | -1.2 | -1.3 | ●-3.8 | ●-10.7 |
| | STMA | mean | 65.2 | 67.1 | 51.3 | 63.8 | 74.8 | 70.8 | 69.5 | 64.4 |
| | | std | 10.0 | 7.7 | 19.0 | 9.1 | 4.4 | 7.6 | 7.7 | 12.1 |
| | | t-val | ●-8.5 | ●-5.0 | ●-6.0 | ●-8.6 | -1.3 | -1.4 | ●-4.7 | ●-10.4 |
| USleep | w/o | mean | 59.1 | 55.9 | 48.4 | 66.6 | 72.9 | 68.8 | 61.3 | 63.3 |
| | | std | 10.3 | 8.3 | 8.2 | 9.3 | 6.1 | 8.3 | 11.0 | 11.3 |
| | | t-val | ●-12.1 | ●-8.0 | ●-13.3 | ·-2.8 | -2.2 | ●-4.3 | ●-6.8 | ●-12.1 |
| DeepSleepNet | w/o | mean | 68.1 | 68.6 | 68.7 | 64.9 | 75.8 | 72.7 | 70.9 | 68.2 |
| | | std | 11.3 | 9.9 | 10.3 | 10.3 | 4.5 | 9.1 | 9.1 | 10.7 |
| | | t-val | ·-3.3 | ·-3.1 | 0.5 | ●-5.1 | -0.5 | 1.7 | -2.8 | ●-3.9 |
| | EA | mean | 66.4 | 66.3 | 62.3 | 67.1 | 74.1 | 73.2 | 68.8 | 68.0 |
| | | std | 11.7 | 10.3 | 11.4 | 10.2 | 4.6 | 9.9 | 9.6 | 11.1 |
| | | t-val | ●-5.1 | ·-3.1 | ·-3.6 | -2.3 | ·-4.1 | 2.1 | ·-3.6 | ●-4.1 |
| | STMA | mean | 67.7 | 70.5 | 64.2 | 64.8 | 74.5 | 72.9 | 71.8 | 67.6 |
| | | std | 10.1 | 8.5 | 11.4 | 10.2 | 4.0 | 7.8 | 7.5 | 10.3 |
| | | t-val | ●-4.7 | -1.4 | -2.7 | ●-5.5 | -1.3 | 2.5 | -1.9 | ●-5.8 |
| AttnSleep | w/o | mean | 68.9 | 65.4 | 61.7 | 67.7 | 75.4 | 73.1 | 68.5 | 68.7 |
| | | std | 10.3 | 10.7 | 14.2 | 10.0 | 4.0 | 8.6 | 10.2 | 10.9 |
| | | t-val | -2.6 | ●-4.4 | ·-3.1 | -1.4 | -1.3 | 2.2 | ●-4.3 | -2.6 |
| | EA | mean | 68.6 | 65.5 | 59.6 | 65.7 | 75.4 | 74.2 | 68.6 | 67.9 |
| | | std | 11.7 | 11.1 | 15.3 | 11.1 | 4.5 | 7.1 | 10.5 | 11.9 |
| | | t-val | ·-2.9 | ·-3.4 | ·-3.9 | ●-3.9 | -1.0 | ●4.5 | ·-3.4 | ●-4.0 |
| | STMA | mean | 69.9 | 69.0 | 61.8 | 66.8 | 75.7 | 74.6 | 71.1 | 68.9 |
| | | std | 10.2 | 8.8 | 14.8 | 9.4 | 3.9 | 7.0 | 8.2 | 10.7 |
| | | t-val | -1.7 | -2.9 | ·-3.0 | ·-3.3 | -0.8 | ●5.2 | -2.9 | -2.2 |
| TSMNet | w/o | mean | 68.0 | 65.9 | 66.7 | 63.6 | 73.4 | 68.7 | 68.3 | 66.3 |
| | | std | 11.2 | 9.6 | 12.4 | 11.3 | 3.9 | 9.3 | 8.9 | 11.1 |
| | | t-val | ·-3.7 | ●-6.5 | -1.0 | ●-6.8 | ·-3.4 | ·-3.8 | ●-7.0 | ●-8.2 |
| | EA | mean | 67.5 | 66.7 | 67.0 | 62.1 | 73.3 | 68.1 | 68.7 | 65.5 |
| | | std | 9.9 | 11.2 | 11.1 | 12.0 | 2.2 | 9.7 | 9.8 | 11.2 |
| | | t-val | ●-4.3 | ·-3.4 | -0.8 | ●-8.2 | -3.3 | ●-4.3 | ●-4.2 | ●-9.5 |
| | STMA | mean | 69.9 | 70.7 | 68.2 | 65.0 | 73.4 | 69.2 | 71.5 | 67.6 |
| | | std | 9.6 | 7.7 | 10.6 | 9.0 | 5.1 | 7.9 | 7.1 | 9.2 |
| | | t-val | -1.8 | -1.5 | 0.1 | ●-7.3 | -1.9 | ●-4.8 | -2.4 | ●-7.3 |
| | SPDDSBN | mean | 68.2 | 68.9 | 64.2 | 62.2 | 72.8 | 66.6 | 70.1 | 64.9 |
| | | std | 8.8 | 5.8 | 8.5 | 6.0 | 2.8 | 6.4 | 5.3 | 7.5 |
| | | t-val | ●-6.3 | ●-4.8 | ●-8.6 | ●-15.3 | ·-3.8 | ●-13.4 | ●-6.2 | ●-21.4 |
| | SPDIM(bias) | mean | 71.0 | 72.1 | 68.1 | 68.6 | 76.7 | 71.6 | 73.5 | 69.9 |
| | | std | 9.6 | 8.0 | 9.8 | 8.5 | 3.4 | 6.9 | 7.2 | 8.6 |
| | | t-val | - | - | - | - | - | - | - | - |

Table A2: **Sleep-staging results per dataset.** Summary statistics of the test-set scores (balanced accuracy; higher is better) across public sleep staging datasets. Parameters that were adapted to the test-data with the IM loss are indicated in brackets. Permutation-paired t-tests were used to identify significant differences between our *proposed* (i.e., TSMNet+SPDIM(bias)) and baseline methods (1e4 permutations, 10 tests, t-max correction). Student's t values summarize the effect strength. Significant differences are highlighted ($\cdot$ $p \leq 0.05$, $\bullet$ $p \leq 0.01$, $\bullet$ $p \leq 0.001$).

| Alignment / Mean | $\mathcal{L}_{\text{IM}}$ / $\Theta_{\text{IM}}$ | Dataset: Group: | CAP patient (n=82) | Dreem healthy (n=22) | Dreem patient (n=50) | HMC patient (n=154) | ISRUC healthy (n=10) | ISRUC patient (n=108) | Overall healthy (n=32) | Overall patient (n=394) |
|---|---|---|---|---|---|---|---|---|---|---|
| $m_\phi$ (19) / per domain $j$ | ✓ / $\Phi_j$ | mean | **71.0** | **72.1** | 68.1 | **68.6** | **76.7** | **71.6** | **73.5** | **69.9** |
| | | std | 9.6 | 8.0 | 9.8 | 8.5 | 3.4 | 6.9 | 7.2 | 8.6 |
| | | t-val | - | - | - | - | - | - | - | - |
| $m_\phi^\#$ (20) per domain $j$ | ✓ / $\varphi_j$ | mean | 68.6 | 70.0 | 66.8 | 64.7 | 74.6 | 68.3 | 71.4 | 66.8 |
| | | std | 9.2 | 6.6 | 9.0 | 7.1 | 3.2 | 6.4 | 6.1 | 7.8 |
| | | t-val | ● -6.4 | • -3.9 | • -3.7 | ● -12.2 | · -3.9 | ● -10.0 | ● -5.3 | ● -16.8 |
| $\tilde{m}_\phi$ (14) per domain $j$ | ✗ / - | mean | 68.2 | 68.9 | 64.2 | 62.2 | 72.8 | 66.6 | 70.1 | 64.9 |
| | | std | 8.8 | 5.8 | 8.5 | 6.0 | 2.8 | 6.4 | 5.3 | 7.5 |
| | | t-val | ● -6.3 | ● -4.8 | ● -8.6 | ● -15.3 | · -3.8 | ● -13.4 | ● -6.2 | ● -21.4 |
| $\tilde{m}_\phi$ (14) global | ✗ / - | mean | 68.0 | 65.9 | 66.7 | 63.6 | 73.4 | 68.7 | 68.3 | 66.3 |
| | | std | 11.2 | 9.6 | 12.4 | 11.3 | 3.9 | 9.3 | 8.9 | 11.1 |
| | | t-val | • -3.7 | ● -6.5 | -1.0 | ● -6.8 | · -3.4 | ● -3.8 | ● -7.0 | ● -8.2 |
| $\tilde{m}_\phi$ (14) per domain $j$ | ✓ / bias in $\psi$ | mean | 69.8 | 70.3 | 65.7 | 64.1 | 74.7 | 68.9 | 71.7 | 66.8 |
| | | std | 9.5 | 6.8 | 9.2 | 6.9 | 2.7 | 6.8 | 6.1 | 8.1 |
| | | t-val | • -3.3 | • -4.0 | ● -7.4 | ● -13.0 | · -3.9 | ● -8.1 | ● -5.4 | ● -15.6 |
| $\tilde{m}_\phi$ (14) per domain $j$ | ✓ / $\psi$ | mean | 67.4 | 68.8 | 63.1 | 62.3 | 73.2 | 67.7 | 70.2 | 64.9 |
| | | std | 10.5 | 8.5 | 10.9 | 9.8 | 5.0 | 8.8 | 7.7 | 10.1 |
| | | t-val | ● -5.9 | · -3.1 | ● -6.1 | ● -14.5 | · -3.8 | ● -8.6 | ● -4.3 | ● -18.0 |
| $\tilde{m}_\phi$ (14) per domain $j$ | ✓ / $\theta \cup \psi$ | mean | 54.8 | 46.9 | 36.2 | 35.8 | 53.6 | 45.9 | 49.0 | 42.5 |
| | | std | 14.8 | 13.8 | 11.4 | 11.2 | 13.4 | 12.5 | 13.8 | 14.5 |
| | | t-val | ● -13.3 | ● -9.6 | ● -17.1 | ● -32.8 | ● -6.4 | ● -23.0 | ● -11.6 | ● -39.9 |

Table A3: **Sleep-staging ablation study results per dataset.** Summary statistics (mean, std, t-val) of the test-set scores (balanced accuracy; higher is better) across public sleep staging datasets. All statistics are computed at the subject level. Permutation-paired t-tests were used to identify significant differences (1e4 permutations, 6 tests, t-max correction). Significant differences are highlighted (· $p \leq 0.05$, • $p \leq 0.01$, ● $p \leq 0.001$). For a summary across datasets and groups see Table 3 in the main manuscript.

## A.5 EEG-BASED MOTOR IMAGERY BCI

| | | | session | | | | | subject | | | | |
|---|---|---|---|---|---|---|---|---|---|---|---|---|
| **Model** | **SFUDA** | Evaluation : Dataset : | 2014001 (n=9) | 2014004 (n=9) | 2015001 (n=12) | Zhou2016 (n=4) | Overall (n=34) | 2014001 (n=9) | 2014004 (n=9) | 2015001 (n=12) | Zhou2016 (n=4) | Overall (n=34) |
| EEGNet v4 | w/o | mean | 41.1 | 74.5 | 72.4 | 71.7 | 64.6 | 42.6 | 69.0 | 61.4 | 72.6 | 59.7 |
| | | std | 17.9 | 14.6 | 16.3 | 5.8 | 20.6 | 16.6 | 10.7 | 8.3 | 3.7 | 15.6 |
| | | t-val | · -4.8 | 0.2 | · -4.1 | -4.4 | ● -5.0 | · -4.3 | -1.7 | -2.3 | -3.4 | ● -4.2 |
| | EA | mean | 53.2 | 75.3 | 76.0 | 77.2 | 69.9 | 48.6 | 72.0 | 70.7 | 70.6 | 65.2 |
| | | std | 21.8 | 15.4 | 18.3 | 2.9 | 19.7 | 17.2 | 11.1 | 14.7 | 6.8 | 16.7 |
| | | t-val | • -5.2 | 0.8 | -2.7 | -8.9 | ● -4.4 | -1.9 | -0.1 | -0.5 | -3.2 | -1.5 |
| | STMA | mean | 28.9 | 73.8 | 55.5 | 67.1 | 54.7 | 50.0 | 71.7 | 68.5 | 73.1 | 65.0 |
| | | std | 5.5 | 14.4 | 9.5 | 14.5 | 20.2 | 16.1 | 12.4 | 14.3 | 2.2 | 16.0 |
| | | t-val | • -7.3 | -0.2 | • -8.3 | -2.4 | ● -6.8 | -1.6 | -0.2 | -0.8 | -9.1 | -1.5 |
| EEG-Conformer | w/o | mean | 48.6 | 75.0 | 71.6 | 73.9 | 66.7 | 41.7 | 69.4 | 60.6 | 74.6 | 59.6 |
| | | std | 17.0 | 11.4 | 18.5 | 3.8 | 18.4 | 15.2 | 10.9 | 11.7 | 6.0 | 16.6 |
| | | t-val | • -6.1 | 0.5 | · -4.0 | -6.2 | ● -5.5 | · -4.1 | -1.7 | -2.4 | -2.8 | ● -4.1 |
| ATCNet | w/o | mean | 52.2 | 74.7 | 72.8 | 80.3 | 68.7 | 41.6 | 68.4 | 59.8 | 67.5 | 58.2 |
| | | std | 18.7 | 13.8 | 16.3 | 4.3 | 18.1 | 16.3 | 10.9 | 8.7 | 8.7 | 15.5 |
| | | t-val | -4.0 | 0.3 | · -3.8 | -2.3 | ● -4.5 | · -4.8 | -3.1 | -2.9 | -3.9 | ● -5.3 |
| EEGInceptionMI | w/o | mean | 47.7 | 71.5 | 72.6 | 75.1 | 66.0 | 39.4 | 67.0 | 59.2 | 63.1 | 56.5 |
| | | std | 17.0 | 12.1 | 15.8 | 9.4 | 18.0 | 13.8 | 9.5 | 9.5 | 6.4 | 14.8 |
| | | t-val | • -6.1 | -1.2 | • -4.9 | -1.9 | ● -6.1 | -3.5 | -2.3 | -2.5 | -6.9 | ● -4.9 |
| TSMNet | w/o | mean | 68.6 | 74.8 | 82.7 | 81.0 | 76.7 | 41.1 | 69.5 | 61.1 | 77.8 | 60.0 |
| | | std | 12.9 | 11.1 | 13.4 | 2.5 | 12.8 | 12.7 | 9.6 | 10.9 | 2.6 | 16.2 |
| | | t-val | -0.5 | 0.4 | -0.6 | -1.8 | -0.9 | · -4.3 | -1.5 | -2.0 | -1.3 | ● -3.5 |
| | EA | mean | 67.2 | 76.0 | 83.6 | 81.9 | 77.0 | 50.0 | 73.0 | 68.2 | 73.3 | 65.3 |
| | | std | 13.7 | 12.0 | 9.5 | 5.2 | 12.6 | 16.9 | 10.6 | 13.7 | 4.3 | 15.9 |
| | | t-val | -1.2 | 2.1 | -0.1 | -0.9 | -0.6 | -1.4 | 0.5 | -0.9 | -2.4 | -1.5 |
| | STMA | mean | 69.9 | 74.7 | 83.0 | 83.3 | 77.3 | 50.2 | 71.9 | 66.9 | 76.8 | 65.0 |
| | | std | 16.0 | 13.1 | 7.7 | 6.3 | 12.7 | 17.5 | 11.5 | 13.0 | 5.5 | 16.0 |
| | | t-val | 0.1 | 0.4 | -0.3 | -0.4 | -0.2 | -1.4 | -0.2 | -1.2 | -1.2 | -1.7 |
| | IM (classifier bias) | mean | 69.9 | 74.2 | 83.2 | 78.3 | 76.7 | 52.9 | 70.6 | 73.0 | 75.4 | 67.3 |
| | | std | 17.5 | 12.5 | 10.1 | 3.0 | 13.3 | 15.0 | 9.9 | 15.1 | 3.8 | 15.3 |
| | | t-val | 0.2 | -0.0 | -0.6 | -11.2 | -1.7 | 0.4 | -1.0 | -0.7 | -1.7 | -1.7 |
| | IM (classifier) | mean | 66.6 | 70.9 | 78.2 | 77.0 | 73.0 | 50.5 | 67.0 | 69.7 | 69.0 | 63.8 |
| | | std | 17.2 | 9.5 | 6.6 | 1.7 | 11.5 | 14.9 | 9.1 | 12.4 | 7.5 | 14.1 |
| | | t-val | -3.1 | -3.0 | · -3.8 | -17.2 | ● -6.8 | -1.1 | · -4.1 | · -3.9 | -2.9 | ● -5.0 |
| | IM (all) | mean | 37.9 | 50.9 | 70.8 | 55.4 | 55.0 | 36.0 | 50.4 | 63.2 | 56.0 | 51.8 |
| | | std | 10.9 | 1.7 | 7.8 | 6.4 | 15.1 | 7.5 | 1.7 | 8.3 | 4.1 | 12.5 |
| | | t-val | • -6.6 | • -6.1 | · -4.0 | -7.5 | ● -9.5 | · -4.0 | • -5.2 | -2.5 | -10.3 | ● -7.3 |
| | SPDDSBN | mean | 69.3 | 73.5 | 81.2 | 80.6 | 76.0 | 52.1 | 69.6 | 71.5 | 76.2 | 66.4 |
| | | std | 16.2 | 11.0 | 8.6 | 2.4 | 12.0 | 13.9 | 9.8 | 13.5 | 3.1 | 14.5 |
| | | t-val | -0.3 | -0.9 | -2.4 | -7.0 | -2.8 | -0.3 | -1.9 | -2.6 | -1.9 | · -3.2 |
| | SPDIM (geodesic) | mean | 70.0 | 74.7 | 83.7 | 82.6 | 77.6 | 52.3 | 70.9 | 73.4 | 78.4 | 67.7 |
| | | std | 16.5 | 11.9 | 10.3 | 3.5 | 13.1 | 15.1 | 10.9 | 15.2 | 3.6 | 16.0 |
| | | t-val | 0.4 | 0.8 | 0.1 | -1.4 | 0.1 | -0.3 | -1.4 | -0.3 | -1.1 | -1.6 |
| | SPDIM (bias) | mean | 69.7 | 74.3 | 83.6 | 84.1 | 77.5 | 52.6 | 72.2 | 73.6 | 80.4 | 68.5 |
| | | std | 18.4 | 12.2 | 10.6 | 2.2 | 13.9 | 16.5 | 12.0 | 14.8 | 2.4 | 16.6 |
| | | t-val | - | - | - | - | - | - | - | - | - | - |

*(label ratio = 0.2)*

Table A4: **Motor imagery BCI results for a label ratio of 0.2.** Average and standard deviation of test-set scores (balanced accuracy; higher is better) across public motor imagery BCI datasets. For all IM and SPDIM variants, the parameters that were tuned to the test-data with the IM loss are indicated in brackets. Permutation-paired t-tests were used to identify significant differences between the *proposed* (i.e., TSMNet+SPDIM(bias)) and baseline methods (1e4 permutations, 14 tests, t-max correction). Student's t values summarize the effect strength. Significant differences are highlighted (· $p \leq 0.05$, • $p \leq 0.01$, ● $p \leq 0.001$).

| | | | session | | | | | subject | | | | |
|---|---|---|---|---|---|---|---|---|---|---|---|---|
| | Evaluation: | | 2014001 | 2014004 | 2015001 | Zhou2016 | **Overall** | 2014001 | 2014004 | 2015001 | Zhou2016 | **Overall** |
| **Model** | **SFUDA** | Dataset: | (n=9) | (n=9) | (n=12) | (n=4) | (n=34) | (n=9) | (n=9) | (n=12) | (n=4) | (n=34) |
| EEGNet v4 | w/o | mean | 41.0 | 73.1 | 73.5 | 70.6 | 64.4 | 43.6 | 69.6 | 61.3 | 75.1 | 60.4 |
| | | std | 16.6 | 15.0 | 15.8 | 6.5 | 20.3 | 16.7 | 8.1 | 8.8 | 3.2 | 15.4 |
| | | t-val | • -6.5 | · -3.6 | • -4.7 | -3.8 | ● -6.3 | ·· -5.0 | -2.1 | -2.4 | -2.8 | ● -4.4 |
| | EA | mean | 54.5 | 76.3 | 75.7 | 79.6 | 70.7 | 49.9 | 74.0 | 72.5 | 73.9 | 67.1 |
| | | std | 19.9 | 15.5 | 17.1 | 2.9 | 18.7 | 16.9 | 10.4 | 14.2 | 3.7 | 16.6 |
| | | t-val | • -6.3 | -1.0 | · -3.7 | -4.9 | ● -5.5 | -3.4 | 1.5 | -0.3 | -4.8 | -1.2 |
| | STMA | mean | 30.1 | 74.4 | 55.9 | 68.3 | 55.4 | 49.7 | 72.6 | 69.9 | 75.0 | 65.9 |
| | | std | 4.7 | 15.8 | 8.0 | 16.8 | 20.1 | 16.9 | 10.6 | 14.6 | 2.4 | 16.4 |
| | | t-val | • -9.4 | -1.8 | • -9.9 | -1.8 | ● -7.6 | ·· -3.7 | -0.8 | -0.7 | -4.9 | -1.6 |
| EEG-Conformer | w/o | mean | 46.9 | 73.8 | 73.1 | 71.1 | 66.1 | 42.6 | 68.8 | 60.1 | 74.5 | 59.5 |
| | | std | 17.2 | 12.8 | 17.2 | 3.2 | 18.7 | 16.7 | 10.0 | 10.7 | 4.4 | 16.1 |
| | | t-val | • -8.9 | · -3.6 | • -4.5 | -8.5 | ● -7.2 | ·· -5.0 | -2.8 | -2.6 | -4.1 | ● -4.7 |
| ATCNet | w/o | mean | 51.6 | 74.3 | 73.2 | 77.9 | 68.3 | 42.7 | 68.5 | 60.2 | 69.8 | 58.9 |
| | | std | 18.4 | 13.8 | 15.6 | 4.3 | 17.8 | 16.4 | 8.9 | 8.4 | 6.3 | 14.9 |
| | | t-val | • -5.6 | -3.0 | • -4.7 | -3.0 | ● -6.3 | • -5.7 | · -3.6 | -2.8 | -4.6 | ● -5.5 |
| EEGInc-eptionMI | w/o | mean | 49.1 | 71.7 | 73.1 | 72.3 | 66.3 | 39.7 | 67.3 | 59.5 | 67.6 | 57.3 |
| | | std | 17.9 | 12.6 | 15.7 | 8.5 | 17.7 | 12.7 | 8.1 | 9.2 | 5.4 | 14.6 |
| | | t-val | • -7.3 | · -3.8 | • -5.6 | -2.6 | ● -7.7 | ·· -5.1 | -2.8 | -2.5 | -5.2 | ● -5.2 |
| TSMNet | w/o | mean | 69.7 | 75.1 | 82.2 | 80.0 | 76.8 | 43.0 | 68.0 | 61.7 | 77.5 | 60.3 |
| | | std | 11.8 | 11.2 | 12.9 | 2.0 | 12.1 | 13.3 | 9.3 | 11.4 | 3.7 | 15.6 |
| | | t-val | -2.6 | -2.4 | -2.0 | -5.3 | ● -4.6 | • -6.0 | -3.2 | -2.0 | -1.7 | ·· -3.9 |
| | EA | mean | 71.1 | 77.1 | 85.2 | 83.8 | 79.2 | 51.2 | 73.1 | 72.5 | 75.0 | 67.3 |
| | | std | 13.3 | 12.8 | 9.7 | 2.6 | 12.2 | 15.1 | 11.1 | 13.6 | 1.8 | 15.6 |
| | | t-val | -2.1 | -0.5 | -0.0 | -1.0 | -1.8 | -1.6 | -0.6 | -0.3 | -8.1 | -1.1 |
| | STMA | mean | 71.9 | 76.3 | 84.9 | 82.8 | 78.9 | 52.5 | 72.5 | 70.1 | 77.1 | 66.9 |
| | | std | 15.2 | 13.3 | 9.2 | 3.0 | 12.6 | 16.4 | 11.3 | 14.2 | 1.9 | 15.6 |
| | | t-val | -1.6 | -2.4 | -0.3 | -4.3 | -2.0 | -1.7 | -1.3 | -0.7 | -4.0 | -1.3 |
| | SPDDSBN | mean | 73.7 | 77.4 | 85.3 | 84.6 | 80.0 | 54.6 | 73.3 | 74.3 | 80.9 | 69.6 |
| | | std | 15.3 | 12.5 | 10.4 | 2.3 | 12.5 | 16.1 | 11.1 | 14.7 | 1.5 | 15.9 |
| | | t-val | - | - | - | - | - | - | - | - | - | - |

label ratio = 1.0

Table A5: **Motor imagery BCI results for balanced data (i.e., label ratio of 1.0).** Average and standard deviation of test-set scores (balanced accuracy; higher is better) across public motor imagery BCI datasets. For all IM and SPDIM variants, the parameters that were tuned to the test-data with the IM loss are indicated in brackets. Permutation-paired t-tests were used to identify significant differences between the *TSMNet+SPDDSBN* and baseline methods (1e4 permutations, 9 tests, t-max correction). Student's t values summarize the effect strength. Significant differences are highlighted ($\cdot$ $p \leq 0.05$, $\bullet$ $p \leq 0.01$, $\bullet$ $p \leq 0.001$).

### A.6    SIMULATIONS

#### A.6.1    IMPLEMENTATION DETAILS

We generated covariance matrices $C_i \in \mathcal{S}_P^+$ with $P = 2$. To do so, we first generated log-space features $s_i \mathbb{R}^{P(P+1)/2}$, defined in (11), using the scikit-learn function `make_classification` with 2 dimensions encoding label information. The data were then normalized to have zero mean and unit variance. To obtain $E_i \in \mathcal{S}_P^+$, we applied $\mathrm{upper}^{-1}$ and $\mathrm{Exp}_{I_P}$, as defined in (5). At this level, the data were split across source domains and the target domain, with each domain receiving 500 observations. Finally, the data were projected to the channel space using domain-specific mixing matrices $A_j$, as defined in (9). To introduce label shifts, we artificially varied the label ratio (LR), defined as the proportion of the minority class to the majority class, in the target domain via randomly dropping samples.

We used balanced accuracy as evaluation metric and examined the performance of SPDIM(bias) and SPDIM(geodesic) against RCT (Zanini et al., 2017) over different label ratios. Figure 2 summarizes the results for different SNR levels. To control the SNR, we varied the class separability parameter of the `make_classification` function. Additional Figures summarize the methods' performance over parameters $|\mathcal{J}_s|$ (Figure A1), $M_j$ (Figure A2), $P$ (Figure A3), and $D$ (Figure A4).

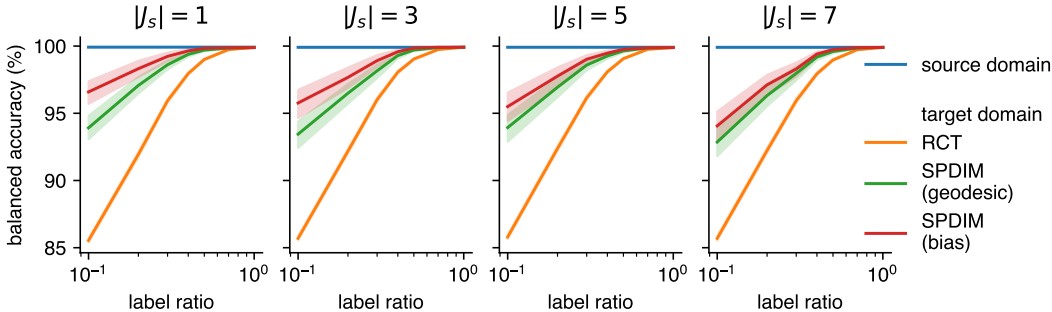

Figure A1: **Performance over the number of source domains.** Same parameters as in Figure 2 (panel 3) but for a different number of source domains $|\mathcal{J}_s| \in \{1, 3, 5, 7\}$.

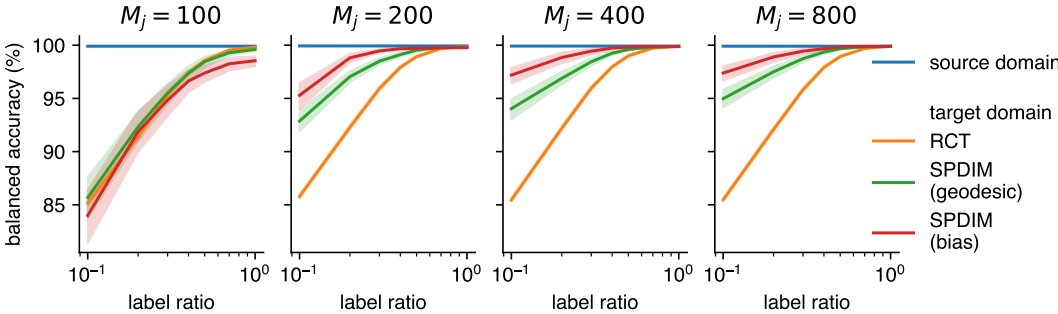

Figure A2: **Performance over the number of samples per domain** $M_j$**.** Same as Figure 2 (panel 3) but for a different number of samples per domains $M_j \in \{100, 200, 400, 800\}$.

### A.7    IMPLEMENTATION DETAILS

#### A.7.1    TSMNET

We used the TSMNet as provided in the public reference implementation as follows:

**Architecture** The feature extractor $f_\theta$ has two convolutional layers, followed by covariance pooling (Acharya et al., 2018), BiMap (Huang & Gool, 2017), and ReEig Huang & Gool (2017) layers. The first convolutional layer operates convolution along the temporal dimension, implementing a

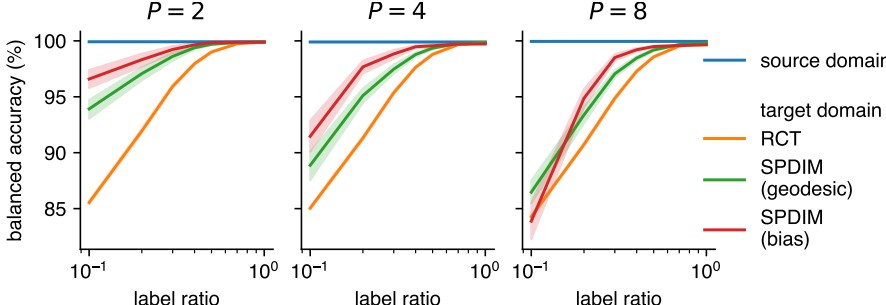

Figure A3: **Performance over the number of dimensions $P$.** Same parameters as in Figure 2 (panel 3) but for a different number of dimensions $P \in \{2, 4, 8\}$.

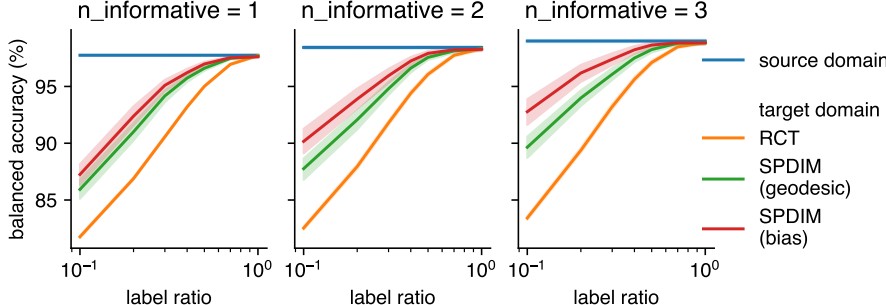

Figure A4: **Performance over the number of informative sources.** Same as Figure 2 (panel 3) but for a different number of informative dimensions encoding label information in $s$, as defined in (11). The `n_informative` parameter of the scikit-learn function `make_classification` effectively defines the dimensionality of the label encoding subspace $D$.

finite impulse response (FIR) filter bank (4 filters) with learnable parameters. The second convolutional layer applies spatio-spectral filters (40 filters) along the spatial and convolutional channel dimensions. Covariance pooling is then applied along the temporal dimension. A subsequent BiMap layer projects covariance matrices to a D-dimensional subspace (D-20) via bilinear mapping. Next, a ReEig layer rectifies all eigenvalues lower than a threshold $10^{-4}$. We varied the alignment and tangent space mapping layer $m_\phi$ as specified in the main text. Finally, the classification head $g_\psi$ is parametrized as a linear layer with softmax activations.

**Parameter estimation** We used the cross-entropy loss as the training objective, employing the PyTorch framework (Paszke et al., 2019) with extensions for structured matrices (Ionescu et al., 2015) and manifold-constrained gradients (Absil et al., 2008) to propagate gradients through the layers. We stick to the hyper-parameters as provided in the public reference implementation. Specifically, gradients were estimated using fixed-size mini-batches (50 observations; 10 per domain across 5 domains) and updated parameters with the Riemannian ADAM optimizer (Bécigneul & Ganea, 2018) ($10^{-3}$ learning rate, $10^{-4}$ weight decay, $\beta_1 = 0.9$, $\beta_2 = 0.999$). We split the source domains' data into training and validation sets (80% / 20% splits, randomized, stratified by domain and label) and iterated through the training set for 100 epochs using exhaustive minibatch sampling. After training, the model with minimal loss on the validation data was selected.

### A.7.2 SLEEP STAGING MODEL

We considered four baseline deep learning architectures Chambon (Chambon et al., 2018), Usleep (Perslev et al., 2021), DeepSleepNet (Supratak et al., 2017), and AttnNet (Eldele et al., 2023) here. Although DeepSleepNet and AttnNet are initial proposed for a single-channel EEG data, there are many related studies (Guillot et al., 2020; Ji et al., 2023; Ma et al., 2024; Guillot & Thorey, 2021) use the model proposed for single-channel data as a baseline for multi-channels data. We use the implementation provided in braindecode (Schirrmeister et al., 2017) for all architectures above, and stick to all model hyper-parameters as provided in the braindecode. We used similar learning-related hyper-parameters with an Adam optimizer (e.g., early stopping, no LR scheduler, same batch size,

similar number of epochs) to TSMNet A.7.1. We split the source domains' data into training and validation sets (80% / 20% splits, randomized, stratified by domain and label), and the model with minimal loss on the validation data was selected after training.

