# OpenReview forum: "SPDIM: Source-Free Unsupervised Conditional and Label Shift Adaptation in EEG"
_ICLR.cc/2025/Conference — ICLR 2025 Poster_

### Official Review · Reviewer_GP1Q · 2024-10-30

**Soundness:** 3
**Presentation:** 3
**Contribution:** 2
**Rating:** 6
**Confidence:** 3

**Summary:**

Due to the difficulty of the SPD method in handling label shift issues, this paper proposes a geometric deep learning framework, SPDIM, for SFUDA problems under specific distribution shifts, including label shifts. SPDIM employs the information maximization principle to learn a single SPD-manifold-constrained parameter per target domain. Using public EEG-based brain-computer interface and sleep staging datasets, we demonstrate that SPDIM outperforms prior approaches.

**Strengths:**

This paper investigates the label shift problem in SFUDA, with a very strong motivation and significant practical relevance.

The method proposed in this paper has a certain theoretical foundation, and the derivation of some propositions may provide inspiration for solving the label shift problem.

**Weaknesses:**

**The clarifications and revisions have addressed most of my concerns.**

---

Although this paper focuses on EEG SFUDA problems, the proposed method does not appear to be specifically designed for EEG but seems to be a more general approach applicable to any label shift scenario. From the perspective of EEG research, the method lacks specificity for EEG data, while from the perspective of SFUDA research, the paper only validates the method on EEG data, lacking more reliable experimental verification.

The experiments are not solid. The paper does not clearly present the experimental setup, such as the hyperparameters of the models, the partitioning method of the source and target domains, etc. Additionally, the EEG decoding methods compared in the experiments are not sufficiently strong. The paper does not compare some classic EEG decoding models, such as EEGNet and EEG Conformer, nor does it compare some sleep staging models, such as DeepSleepNet. The domain adaptation methods only compare Information Maximization (IM), and such insufficient comparisons are not enough to prove the superiority of the proposed method.

The writing of this paper still has some room for improvement. For example: Figure 1 has low resolution, and the four sub-figures in Figure 2 lack sub-titles.

**Questions:**

Please see the weakness.

---

> ### Author Response · Authors · 2024-11-21
> **Response [1/2] to Official Review by Reviewer GP1Q - Weaknesses**
>
> Thank you very much for your effort to assess our submission and the provided feedback.
> For your convenience, we decided to include copies of the revised manuscript [manuscript_revised.pdf](https://anonymous.4open.science/r/SPDIM-ICLR2025--B213/manuscript_revised.pdf) and another file [manuscript_diff_submitted_revised.pdf](https://anonymous.4open.science/r/SPDIM-ICLR2025--B213/manuscript_diff_submitted_revised.pdf) that highlights the changes compared to the submitted manuscript in the anonymous repository.
> Please find our answers to your comments below.

---

> ### Author Response · Authors · 2024-11-21
> **Response [2/2] to Official Review by Reviewer GP1Q - Weaknesses**
>
> > **Although this paper focuses on EEG SFUDA problems, the proposed method does not appear to be specifically designed for EEG but seems to be a more general approach applicable to any label shift scenario. From the perspective of EEG research, the method lacks specificity for EEG data, while from the perspective of SFUDA research, the paper only validates the method on EEG data, lacking more reliable experimental verification.**
>
> Thank you for sharing your opinion.
> We used the poor generalization of EEG-based neurotechnology as a motivation for our framework.
> Although the method is generally applicable to any data that follows our generative model, we based our assumptions and theoretical considerations to EEG data.
> Consequently, we decided to evaluate our framework with EEG data and keep the submission focused to EEG, as clearly indicated in the title, abstract and the introduction.
>
> > **The experiments are not solid. The paper does not clearly present the experimental setup, such as the hyperparameters of the models, the partitioning method of the source and target domains, etc.**
>
> We apologize for failing to fully convey the implementation details behind our approach in the submitted manuscript.
> We included implementation details (appendix 6) about model hyperparameters and experiment settings. Additionally, we aimed to improve clarity of the revised manuscript by rewriting sections 3 and 4, and creating a new overview Figure.
>
> > **Additionally, the EEG decoding methods compared in the experiments are not sufficiently strong. The paper does not compare some classic EEG decoding models, such as EEGNet and EEG Conformer, nor does it compare some sleep staging models, such as DeepSleepNet.**
>
> Thank you for sharing your concern.
> We compared our approach to two recent deep learning models that were specifically proposed for sleep staging: Usleep and Chambon.
> We decided to exclude DeepSleepNet because it is designed for single-channel EEG data, while we considered multi-channel datasets.
> For comparisons between EEGNet and TSMNet architectures on motor imagery datasets, please refer to (Kobler+2022,*NeurIPS*).
>
> *References*:
>
> - R. Kobler, J. Hirayama, Q. Zhao, and M. Kawanabe, “SPD domain-specific batch normalization to crack interpretable unsupervised domain adaptation in EEG,” in Advances in Neural Information Processing Systems, S. Koyejo, S. Mohamed, A. Agarwal, D. Belgrave, K. Cho, and A. Oh, Eds., Curran Associates, Inc., 2022, pp. 6219-6235. url: https://proceedings.neurips.cc/paper_files/paper/2022/file/28ef7ee7cd3e03093acc39e1272411b7-Paper-Conference.pdf
>
> > **The domain adaptation methods only compare Information Maximization (IM), and such insufficient comparisons are not enough to prove the superiority of the proposed method.**
>
> Indeed, the feedback from all reviewers indicated that we should compare our framework with more baseline methods. Following this request, we decided to include additional multi-source SFUDA methods, including, EA (He&Wu2019,*IEEE TBME*) and STMA (Gnassounou+2024,*arXiv*). We additionally changed the wording to better delineate the difference between the previously proposed SPDDSBN (Kobler+2022,*NeurIPS*) method and our proposed SPDIM framework.
> To keep the comparison focused, we decided to exclude semi-supervised as well as single source/target domain UDA methods.
>
> *References*:
>
> - H. He and D. Wu, “Transfer Learning for Brain-Computer Interfaces: A Euclidean Space Data Alignment Approach,” IEEE Trans. Biomed. Eng., vol. 67, no. 2, pp. 399-410, Feb. 2020, doi: 10.1109/TBME.2019.2913914.
>
> - T. Gnassounou, A. Collas, R. Flamary, K. Lounici, and A. Gramfort, “Multi-Source and Test-Time Domain Adaptation on Multivariate Signals using Spatio-Temporal Monge Alignment,” Jul. 19, 2024, arXiv: 2407.14303. url: http://arxiv.org/abs/2407.14303
>
> - R. Kobler, J. Hirayama, Q. Zhao, and M. Kawanabe, “SPD domain-specific batch normalization to crack interpretable unsupervised domain adaptation in EEG,” in Advances in Neural Information Processing Systems, S. Koyejo, S. Mohamed, A. Agarwal, D. Belgrave, K. Cho, and A. Oh, Eds., Curran Associates, Inc., 2022, pp. 6219-6235. url: https://proceedings.neurips.cc/paper_files/paper/2022/file/28ef7ee7cd3e03093acc39e1272411b7-Paper-Conference.pdf
>
> > **The writing of this paper still has some room for improvement. For example: Figure 1 has low resolution, and the four sub-figures in Figure 2 lack sub-titles.**
>
> Thank you for reporting these back to us.
> We apologize for introducing oversight in the submitted manuscript.
> To improve clarity, we created an updated Figure 1 and added titles to the sub-figures in Figure 2.
> As indicated in the caption of Figure 2, the sub-figures summarize the results for different cases of class separability.

---

> > ### Comment · Reviewer_GP1Q · 2024-11-21
> > **Response to the rebuttal**
> >
> > Thank you for the detailed responses and the improvements made to the revised manuscript. While the paper has shown some improvement, particularly in the clarity of experimental setups and the visual quality of figures, several key issues remain unresolved. Specifically:
> >
> > 1. **Lack of specificity to EEG data**
> >    The motivation of this work is stated to address the generalization challenges in EEG research. However, label shift and generalization problems are prevalent across many domains. Although the theoretical assumptions and analyses are grounded in EEG data, the proposed method itself does not incorporate any specific design tailored to the unique characteristics of EEG data, as acknowledged in the rebuttal. This limits the contribution of this work to the EEG community. Additionally, by validating the method only on EEG data, its broader applicability to domain adaptation research is not sufficiently demonstrated.
> >
> > 2. **Insufficient comparative experiments**
> >    The comparative experiments presented in the paper are insufficient to demonstrate the proposed method's contributions:
> >    - For motor imagery tasks, the EEG community already has numerous well-established backbone models, such as EEGNet, EEG-Conformer, and ATCNet. However, the proposed method is demonstrated only with TSMNet, without comparisons to these established models, making it difficult to evaluate its relative performance and advantages.
> >    - For sleep staging tasks, while DeepSleepNet is designed for single-channel data, there are other multi-channel models such as SeqSleepNet and XSleepNet. The lack of comparisons with these relevant models further reduces the strength of the evaluation.
> >
> > 3. **Limited domain contributions**
> >    The method does not show specific contributions to the EEG field, as no design or implementation explicitly leverages EEG-specific characteristics. Moreover, restricting the experiments solely to EEG data limits the work's potential impact and generalizability within the broader domain adaptation research community.
> >
> > Of course, all above only represent the opinion of reviewer GP1Q, and other reviewers’ feedback on the rebuttal will also be taken into account. However, based on the revised manuscript and the rebuttal, reviewer GP1Q has concerns that the contributions of this work are limited, both for the EEG community and for the domain adaptation field.

---

> > > ### Author Response · Authors · 2024-11-27
> > > **Response[1/3] to Response to the rebuttal**
> > >
> > > Thank you very much for providing feedback again. We are deeply sorry that we did not meet your request in our last response; we hope that our efforts here ease your concern.
> > >
> > > > **For motor imagery tasks, the EEG community already has numerous well-established backbone models, such as EEGNet, EEG-Conformer, and ATCNet. However, the proposed method is demonstrated only with TSMNet, without comparisons to these established models, making it difficult to evaluate its relative performance and advantages.**
> > >
> > > Following your request, we conducted additional experiments with classical EEG models for motor imagery, including, EEGNet (Lawhern+2018, *J. Neural Eng.* ), EEG-Conformer (Song+2022, *IEEE TNSRE*), ATCNet (Altaheri+2022, *IEEE Trans. Ind.Inform.*), and EEGInceptionMI (Zhang+2021, *J. Neural Eng.*).
> > >
> > > Moreover, we followed (Xu+2020, *Front. Hum. Neurosci.*) to combine two multi-source SFUDA methods EA (He&Wu2019,*IEEE TBME*) and STMA (Gnassounou+2024,*arXiv*) with EEGNet.
> > > Note that both multi-source SFUDA methods are model-agnostic techniques that are applied to the EEG data before a classifier is fitted.
> > >
> > > The additional results are summarized in Table A4 (w/ label shifts) and Table A5 (w/o label shifts) in the latest revision of the manuscript.
> > > Comparing the results w/o label shifts (Table A5), we can confirm that TSMNet is a highly competitive architecture for motor imagery. Especially in the inter-session setting, TSMNet clearly outperformed SoA deep learning architectures like EEG-Conformer and ATCNet by a large margin (approx. 10\%).
> > > The margin was slightly smaller in the inter-subject setting for the deep learning SFUDA=w/o baselines (approx. 7\%) and declined to (approx. 2\%) for EEGNet if it was combined with EA or STMA.
> > > Under an additional severe label shift (LR = 0.2) in the target domain, this overall trend continued - see Table A5.
> > > These supplementary results support the effectiveness of SFUDA and the TSMNet architecture for EEG data.
> > >
> > >
> > > *References*:
> > > - Lawhern, Vernon J., et al. "EEGNet: a compact convolutional neural network for EEG-based brain-computer interfaces." Journal of neural engineering 15.5 (2018): 056013.
> > >
> > > - Song, Yonghao, et al. "EEG conformer: Convolutional transformer for EEG decoding and visualization." IEEE Transactions on Neural Systems and Rehabilitation Engineering 31 (2022): 710-719.
> > >
> > > - Altaheri, Hamdi, Ghulam Muhammad, and Mansour Alsulaiman. "Physics-informed attention temporal convolutional network for EEG-based motor imagery classification." IEEE transactions on industrial informatics 19.2 (2022): 2249-2258.
> > >
> > > - Zhang, Ce, Young-Keun Kim, and Azim Eskandarian. "EEG-inception: an accurate and robust end-to-end neural network for EEG-based motor imagery classification." Journal of Neural Engineering 18.4 (2021): 046014.
> > >
> > > - Xu, Lichao, et al. "Cross-dataset variability problem in EEG decoding with deep learning." Frontiers in human neuroscience 14 (2020): 103.
> > >
> > > - H. He and D. Wu, “Transfer Learning for Brain-Computer Interfaces: A Euclidean Space Data Alignment Approach,” IEEE Trans. Biomed. Eng., vol. 67, no. 2, pp. 399-410, Feb. 2020, doi: 10.1109/TBME.2019.2913914.
> > >
> > > - T. Gnassounou, A. Collas, R. Flamary, K. Lounici, and A. Gramfort, “Multi-Source and Test-Time Domain Adaptation on Multivariate Signals using Spatio-Temporal Monge Alignment,” Jul. 19, 2024, arXiv: 2407.14303. url: http://arxiv.org/abs/2407.14303

---

> > > > ### Author Response · Authors · 2024-11-27
> > > > **Response[2/3] to Response to the rebuttal**
> > > >
> > > > > **For sleep staging tasks, while DeepSleepNet is designed for single-channel data, there are other multi-channel models such as SeqSleepNet and XSleepNet. The lack of comparisons with these relevant models further reduces the strength of the evaluation.**
> > > >
> > > > Thank you for pointing us to alternative methods. After some research, we identified baseline implementations of SeqSleepNet (https://github.com/pquochuy/SeqSleepNet) and XSleepNet (https://github.com/pquochuy/xsleepnet). Unfortunately, both are implemented with the TensorFlow framework. Due to a lack of time within the rebuttal period, we could not complete them within the given timeframe.
> > > >
> > > > Still, we wanted to include further baseline methods. Studying recent sleep staging papers carefully, we noticed that many related studies (Guillot+2020,*IEEE TNSRE*; Ji+2023, *IEEE TNSRE*; Ma+2024, *arXiv* ) use the model proposed for single-channel data (e.g., DeepSleepNet) as a baseline for multi-channel data.
> > > > Therefore, we decided to include DeepSleepNet (Supratak+2017, *IEEE TNSRE*) and AttnNet (Eldele+2021, *IEEE TNSRE*) as baseline methods.
> > > >
> > > > Due to the competitive performance of these two models, we decided to also combine them with the multi-source SFUDA methods EA (He&Wu2019,*IEEE TBME*) and STMA (Gnassounou+2024,*arXiv*).
> > > >
> > > > The additional results are summarized in Table 1 and Table A2 in the latest revision of the manuscript.
> > > > Among the published baseline methods (summarized in Table 1) DeeplSeepNet and AttnNet achieve the best overall performance among the considered baseline deep learning methods.
> > > > Still, our proposed TSMNet+SPDIM(bias) method outperformed both approaches.
> > > > While the results for Dreem(healthy) and ISRUC(patient) were numerically slightly lower for our proposed method, the differences were not statistically significant.
> > > >
> > > > Comparing the results in Table A2, we find that the baseline deep learning methods only marginally benefitted from the considered SFUDA methods (EA and STMA), while the proposed SPDIM(bias) significantly increased the performance of TSMNet by (approx. 3.5(patient) to 5(healthy)\%). The other considered SFUDA methods (EA, STMA) yielded only marginal increases. SPDDSBN even lead to a performance drop for patient data.
> > > >
> > > > Altogether, the sleep staging results clearly highlight the effectiveness of our proposed framework.
> > > >
> > > > *References*:
> > > > - Guillot, Antoine, et al. "Dreem open datasets: Multi-scored sleep datasets to compare human and automated sleep staging." IEEE transactions on neural systems and rehabilitation engineering 28.9 (2020): 1955-1965.
> > > >
> > > > - Ji, Xiaopeng, Yan Li, and Peng Wen. "3DSleepNet: A multi-channel bio-signal based sleep stages classification method using deep learning." IEEE Transactions on Neural Systems and Rehabilitation Engineering (2023).
> > > >
> > > > - Ma, Jingying, et al. "ST-USleepNet: A Spatial-Temporal Coupling Prominence Network for Multi-Channel Sleep Staging." arXiv preprint arXiv:2408.11884 (2024).
> > > >
> > > > - Eldele, Emadeldeen, et al. "An attention-based deep learning approach for sleep stage classification with single-channel EEG." IEEE Transactions on Neural Systems and Rehabilitation Engineering 29 (2021): 809-818.
> > > >
> > > > - Supratak, Akara, et al. "DeepSleepNet: A model for automatic sleep stage scoring based on raw single-channel EEG." IEEE transactions on neural systems and rehabilitation engineering 25.11 (2017): 1998-2008.
> > > >
> > > > - H. He and D. Wu, “Transfer Learning for Brain-Computer Interfaces: A Euclidean Space Data Alignment Approach,” IEEE Trans. Biomed. Eng., vol. 67, no. 2, pp. 399-410, Feb. 2020, doi: 10.1109/TBME.2019.2913914.
> > > >
> > > > - T. Gnassounou, A. Collas, R. Flamary, K. Lounici, and A. Gramfort, “Multi-Source and Test-Time Domain Adaptation on Multivariate Signals using Spatio-Temporal Monge Alignment,” Jul. 19, 2024, arXiv: 2407.14303. url: http://arxiv.org/abs/2407.14303

---

> > > > > ### Author Response · Authors · 2024-11-27
> > > > > **Response[3/3] to Response to the rebuttal**
> > > > >
> > > > > > **The motivation of this work is stated to address the generalization challenges in EEG research. However, label shift and generalization problems are prevalent across many domains. Although the theoretical assumptions and analyses are grounded in EEG data, the proposed method itself does not incorporate any specific design tailored to the unique characteristics of EEG data, as acknowledged in the rebuttal. This limits the contribution of this work to the EEG community. Additionally, by validating the method only on EEG data, its broader applicability to domain adaptation research is not sufficiently demonstrated.**
> > > > >
> > > > > Thank you for sharing your opinion.
> > > > > Building upon recent generative statistical models for EEG (Sabbagh+2020, *NeuroImage*; Mello+2023, *Imaging Neuroscience*), we introduce a novel generative model that relaxes the assumptions of prior works - specifically the joint or block diagonizability of the latent source covariance matrices.
> > > > > Hence, we believe that our assumptions and theoretical considerations are firmly grounded within the EEG application domain.
> > > > > Based on our new generative model, we performed theoretical analysis and proposed a method to handle label shifts.
> > > > > Extensive simulations confirmed that our proposed method effectively compensates conditional and label shifts in data sampled from our generative model.
> > > > > Since the empirical EEG data results for SPDIM are highly competitive - even after substantially increasing the considered baseline methods in motor imagery and sleep staging - we now have even stronger evidence that our modeling assumptions are suitable for real EEG data.
> > > > >
> > > > >
> > > > > *References*:
> > > > > - Sabbagh, David, et al. "Predictive regression modeling with MEG/EEG: from source power to signals and cognitive states." NeuroImage 222 (2020): 116893.
> > > > > - Mellot, Apolline, et al. "Harmonizing and aligning M/EEG datasets with covariance-based techniques to enhance predictive regression modeling." Imaging Neuroscience 1 (2023): 1-23.
> > > > >
> > > > > > **The method does not show specific contributions to the EEG field, as no design or implementation explicitly leverages EEG-specific characteristics. Moreover, restricting the experiments solely to EEG data limits the work's potential impact and generalizability within the broader domain adaptation research community.**
> > > > >
> > > > > We appreciate that you shared your opinion.
> > > > > In our opinion, the main contribution of our work is a Riemannian statistical alignment framework for domain adaptation in EEG.
> > > > > We first proposed a realistic EEG generative model, and showed that prior Riemannian statistical alignment approaches like RCT or SPDDSBN hurt generalization under additional label shifts, offering fresh insights into domain adaptation for EEG applications.
> > > > > We then propose a theoretically-based method that can handle conditional and label shifts without requiring labeled target domain data.
> > > > > The experimental results in the latest revised manuscript provide, in our opinion, sufficiently efficent for the effectiveness of our proposed method.

---

> > > > > > ### Comment · Reviewer_GP1Q · 2024-11-27
> > > > > >
> > > > > > Dear Authors,
> > > > > >
> > > > > > Thank you for your thorough responses and the updated version of your paper. Your clarifications and revisions have addressed most of my concerns, and I am pleased to raise my score to 6.
> > > > > >
> > > > > > Best regards,
> > > > > >
> > > > > > Reviewer  GP1Q

---

> > > > > > > ### Author Response · Authors · 2024-11-28
> > > > > > >
> > > > > > > Dear Reviewer GP1Q
> > > > > > >
> > > > > > > Thank you very much for recognizing our work. We are pleased to witness the improvement of our submission based on your thoughtful feedback.
> > > > > > >
> > > > > > > Best regards,
> > > > > > >
> > > > > > > Submission8425 Authors

---

### Official Review · Reviewer_ocKU · 2024-11-01

**Soundness:** 3
**Presentation:** 3
**Contribution:** 3
**Rating:** 6
**Confidence:** 5

**Summary:**

The paper introduces SPDIM, a framework for source-free unsupervised domain adaptation (SFUDA) in EEG-based applications, which are challenged by distribution shifts across sessions or subjects. SPDIM leverages the geometry of symmetric positive definite (SPD) matrices to handle conditional and label shifts, aligning EEG data across domains without requiring labeled target data. The approach introduces a domain-specific SPD-manifold bias to counteract label shifts, and optimizes alignment using an information maximization principle, which prevents mode collapse by ensuring class diversity and prediction confidence. Experimental results on EEG datasets for motor imagery and sleep staging show that SPDIM outperforms baseline SFUDA methods, demonstrating robust generalization across domains even under significant label distribution changes.

**Strengths:**

- The paper is well written.
- The modelisation is original and insightful. I really enjoyed reading the modelisation part of the paper.
- The developed methods are tested on 3 setups: synthetic data, Motor-Imagery and Sleep-Staging.

**Weaknesses:**

- At the time of reviewing the paper, the code is not available: “The repository is not found.” is returned by anonymous.4open.science
- A modelisation per domain of EEG data was proposed in [1] which could be worth citing in your introduction. Indeed, the authors mention there exists a linear mapping per domain to get domain-invariant tangent vectors (and without assumption on the mixing matrix (9)).
- The experiment on motor imagery is limited since you artificialy unbalance the labels. Finding real world data which are naturally unbalanced would add value to the paper.
- The mean accuracy of the 2 proposed methods are within the standard deviation of the recenter for the motor imagery application.
- On the sleep-staging setup, you do not compare with adaptation methods expect recenter. You should compare at least to STMA or TMA (Spatio-Temporal Monge Alignment) which is presented in [2].
- The presentation of the results is not homogeneous between the two applications. In particular, it is strange to me to call an “ablation study” a comparison with other methods.

[1] Collas, Antoine, Rémi Flamary, and Alexandre Gramfort. "Weakly supervised covariance matrices alignment through Stiefel matrices estimation for MEG applications." arXiv preprint arXiv:2402.03345 (2024).

[2] Gnassounou, T., Collas, A., Flamary, R., Lounici, K., & Gramfort, A. (2024). Multi-Source and Test-Time Domain Adaptation on Multivariate Signals using Spatio-Temporal Monge Alignment. arXiv preprint arXiv:2407.14303.


I put a rating of 5 but I am open to increasing it.

**Questions:**

- You mention there are conditional shifts in EEG data (p_j(x|y) changes between domains). Can you relate this with your modelization?
- What is D in the Remark 1?
- Does the Propostion 2 still hold when M_j does not tend to the infinite?
- You train your model on the target domain (in an unsupervised manner). Did you train/test split the target domain?
- How easy to train are the methods you use? e.g. USleep is rarely used as a baseline in other sleep staging papers. Providing infos the lr scheduler, batch size, … would be valuable.
- I am surprised that the spatial covariance is enough to classify sleep stages. Usually, the temporal information is used but not the spatial one. Can you comment on this?

A few typos:
- D and P are both used for the data dimension
- There are “?” in lines 218 and 236.
- Q and U are both used for domain-invariant par of the mixing matrix.

---

> ### Author Response · Authors · 2024-11-21
> **Response [1/4] to Official Review by Reviewer ocKU - Weaknesses [1/2]**
>
> We appreciate the reviewer's effort to provide feedback and suggestions.
> Thank you also for expressing your concerns regarding weaknesses and posing additional questions.
> Please find our detailed responses to the weaknesses and questions below.
>
> For your convenience, we decided to include copies of the revised manuscript [manuscript_revised.pdf](https://anonymous.4open.science/r/SPDIM-ICLR2025--B213/manuscript_revised.pdf) and another file [manuscript_diff_submitted_revised.pdf](https://anonymous.4open.science/r/SPDIM-ICLR2025--B213/manuscript_diff_submitted_revised.pdf) that highlights the changes compared to the sumitted manuscript in the anonymous repository.
>
> ***Weaknesses***
> >
> > **At the time of reviewing the paper, the code is not available: “The repository is not found.” is returned by anonymous.4open.science**
>
> We apologize for the oversight that caused the broken code link at the time of submission.
> We updated the code link in the revised manuscript: https://anonymous.4open.science/r/SPDIM-ICLR2025--B213
>
> > **A modelisation per domain of EEG data was proposed in [1] which could be worth citing in your introduction. Indeed, the authors mention there exists a linear mapping per domain to get domain-invariant tangent vectors (and without assumption on the mixing matrix (9)).**
>
> Thank you very much for sharing this reference; it is indeed very related. We cite it in the revised manuscript.
>
> > **The experiment on motor imagery is limited since you artificially unbalance the labels. Finding real world data which are naturally unbalanced would add value to the paper.**
>
> We understand your concern.
> Having in mind realistic brain-computer interface application settings, the variability of human behavior and environmental factors likely cause label shifts across days and subjects.
> Yet, almost all public motor imagery datasets are generated in a highly controlled lab environment and designed to be balanced.
> To bridge the gap between controlled research settings and real-world scenarios, we decided to include the motor imagery BCI experimental  results in this manuscript.
> To emphasize this demand, we rephrased the motivation for this experiment in the revised manuscript.

---

> ### Author Response · Authors · 2024-11-21
> **Response [2/4] to Official Review by Reviewer ocKU - Weaknesses [2/2]**
>
> > **The mean accuracy of the 2 proposed methods are within the standard deviation of the recenter for the motor imagery application.**
>
> We are deeply sorry to report a typo in the caption of Figure 3, which presents the motor imagery results.
> The error bars shown actually represent the 95% confidence interval of the mean (i.e., standard behavior of the `seaborn` package `barplot` function) rather than the standard deviation.
>
> Knowing that EEG motor imagery performance greatly varies across individual subjects, statistical analyses typically utilize a repeated measures design.
> We decided to compute paired test statistics (specifically paired t-tests) at the subject level.
> The effect strengths (in terms of paired t-values) are summarized in Table A3 in Appendix A.5.
> For example, the paired differences between SPDIM (bias) and SPDDSBN (i.e., RCT in the roginal manuscript) yielded t-values of -2.8 and -3.2 in the cross-session and cross-subject transfer settings.
> These turned out to be significantly different to the distribution under the null hypothesis (i.e., null hypothesis: no difference between SPDIM (bias) and SPDDSBN) that we obtained with permutation testing.
> Hence, while the confidence intervals overlap, we still observed a statistically significant difference between the performance of both methods.
>
>
> > **On the sleep-staging setup, you do not compare with adaptation methods expect recenter. You should compare at least to STMA or TMA (Spatio-Temporal Monge Alignment) which is presented in [2].**
>
> After reading all reviewers' feedback, we noticed that the use of the term RCT might have been misleading. In our framework, we actually use an SPDDSMBN layer (Kobler+2022, *NeurIPS*) to recenter and rescale the data in the latent SPD space. To reduce potential confusion with the RCT method proposed in (Zanini+2017, *IEEE TBME*), we changed the terminology in the revised manuscript.
>
> Thank you also for pointing us to the very recently introduced STMA (/TMA) method.
> We decided to include it as a baseline method for the revised manuscript.
> We based additional sleep-staging experiments on the publicly provided reference implementation (https://github.com/tgnassou/spatio-temporal-monge-alignment) and our evaluation scheme (within dataset; domains correspond to subjects), and can confirm that STMA is a suitable SFUDA approach across models (we considered Chambon and TSMNet).
> The combination of TSMNet+STMA yields the best results among the considered baseline methods (for details, see Table 1 in the revised manuscript). Still, there remains a significant gap to our proposed method (i.e., SPDIM(bias)).
>
> *References*:
>
> - P. Zanini, M. Congedo, C. Jutten, S. Said, and Y. Berthoumieu, “Transfer Learning: A Riemannian Geometry Framework With Applications to Brain-Computer Interfaces,” IEEE Trans. Biomed. Eng., vol. 65, no. 5, pp. 1107-1116, May 2018, doi: 10.1109/TBME.2017.2742541.
>
>
> - R. Kobler, J. Hirayama, Q. Zhao, and M. Kawanabe, “SPD domain-specific batch normalization to crack interpretable unsupervised domain adaptation in EEG,” in Advances in Neural Information Processing Systems, S. Koyejo, S. Mohamed, A. Agarwal, D. Belgrave, K. Cho, and A. Oh, Eds., Curran Associates, Inc., 2022, pp. 6219-6235.
>
> > **The presentation of the results is not homogeneous between the two applications. In particular, it is strange to me to call an “ablation study” a comparison with other methods.**
>
> Thank you for raising this concern. Our intentions with both experiments were slightly different. In the experiment with motor imagery data, we aimed to demonstrate that the SPDIM framework is a useful remedy for conditional label shift compensation frameworks that assume a constant label distribution (e.g., RCT). As a recent representative, we chose the TSMNet architecture which combines end-to-end learning with latent recentering.
> In the sleep staging experiment, we aimed to extend the comparison to relevant baseline methods. Due to public code availability we chose USleep and Chambon. Since multiple reviewers requested additional comparisons to other baseline methods, we decided to include more baseline methods for the sleep staging experiment.
>
> Our intention with the sleep staging ablation study was to indicate that our proposed SPIM framework effectively combines several components introduced by prior works. For example, SPDDSBN (i.e., RCT in the original submission), which was proposed together with the TSMNet architecture, as well as the IM loss.
> As indicated by this reviewer, the choice of presentation might be perceived as strange.
> To improve the presentation, we reorganized Table 2 in the revised manuscript.

---

> ### Author Response · Authors · 2024-11-21
> **Response [3/4] to Official Review by Reviewer ocKU - Questions [1/2]**
>
> > **You mention there are conditional shifts in EEG data (p_j(x|y) changes between domains). Can you relate this with your modelization?**
>
> Thank you for raising this question. In our model we utilize a latent source covariance variable $\mathrm{E} = f_{\mathrm{E}}(y, \mathrm{\varepsilon})$ with deterministic function $f_\mathrm{E}$, defined in (10) and (11) in the original manuscript, and random variables $\mathrm{y} \sim P_{\mathrm{y}}$ and $\mathrm{\varepsilon} \sim P_{\mathrm{\varepsilon}}$.
> Additionally, we model the random variable $\mathrm{z}$ to be zero-mean (i.e., $\mathrm{E} \lbrace z \rbrace = 0$) and its covariance to be defined by $\mathrm{E}$ (i.e., $\mathrm{Cov}(\mathrm{z}) = \mathrm{E}$).
> Finally, we model the observed EEG signals as $\mathrm{x} = f_{\mathrm{x}}(\mathrm{z}, \mathrm{A})$ with deterministic function $f_\mathrm{x}$, defined in (8) in the original manuscript.
> To introduce conditional shifts in $\mathrm{x}$, we model $\mathrm{A}$ to be a function of the domain $j$ (i.e., $\mathrm{A}_j = Q\mathrm{exp}(P_j)$, as defined in (9) in the original manuscript). Consequently, the conditional shifts in the distribution of $\mathrm{x}$ are caused by the domain-specific forward model $A_j$.
>
> We updated Figure 1a in the revised manuscript to graphically summarize our generative model.
>
>
> > **What is D in the Remark 1?**
>
> Thank you for spotting this error; it should be the $B$, as defined in (11) in the original manuscript.
> We apologize for this and any other typos in the submitted version.
> Before submission we streamlined notation with the aim to minimize confusion across symbols but missed to update some occurrences.
> We updated this and other notation errors in the revised manuscript.
>
> > **Does the Proposition 2 still hold when M_j does not tend to the infinite?**
>
> Interesting question. For our proof of proposition 2 (listed in Appendix A.2) we utilized proposition 1, where we utilized $M_j \rightarrow \infty$ in its proof (listed in Appendix A.1). Relaxing or reducing assumptions is definitely an interesting direction for future work.
>
> > **You train your model on the target domain (in an unsupervised manner). Did you train/test split the target domain?**
>
> No, we consider the entire target domain data as the training set, and use the IM loss to adapt the model parameters for a fixed number of epochs.
> We apologize for failing to fully convey the implementation details behind our approach in the submitted manuscript.
> We added this missing piece of information in the revised manuscript by introducing dedicated sub-sections for source domain training and target domain adaptation in section 4. Additionally, we updated Figure 1 to graphically emphasize the separation between source-domain training and the SFUDA with SPDIM.
>
> > **How easy to train are the methods you use? e.g. USleep is rarely used as a baseline in other sleep staging papers. Providing infos the lr scheduler, batch size, … would be valuable.**
>
>
> Thank you for sharing your opinion about USleep.
> Without being experts in sleep staging, we simply picked Usleep because it is a relatively recent method and is available in the `Braindecode` package.
> To reduce the risk of overfitting the architectures to the data, we decided to stick to the model hyper-parameters (e.g., TSMNet, Chambon, USleep) as provided in the public reference implementations.
> To facilitate a somewhat fair comparison, we decided to use similar learning related hyper-parameters (e.g., early stopping, no LR scheduler, same batch size, similar number of epochs).
> Knowing that this could compromise baseline model performance, we additionally verified that our results are comparable to the ones that are reported in the associated papers (for settings with similar datasets and evaluation scenarios).
> We added additional training-related information in an appendix of the revised manuscript.
>
> > **I am surprised that the spatial covariance is enough to classify sleep stages. Usually, the temporal information is used but not the spatial one. Can you comment on this?**
>
> Your intuition is right. Sleep stages are mostly expressed in terms of global power changes in rhythmic brain oscillations. So, an efficient decoder should definitely utilize temporal information.
> Actually, the feature extractor $f_{\theta}$ that comes with the TSMNet architecture combines spatial and temporal convolution layers. Specifically, the first two convolution layers are similar to the ones of ShallowConvNet (Schirrmeister+2017, *Hum. Brain Mapp.*).
> We did not highlight this information in the submitted version. Interpreting the reviewers' comments, we rewrote several parts of the revised manuscript to clarify the role of the feature extractor in our model.
>
> *References*
>
> R. T. Schirrmeister et al., “Deep learning with convolutional neural networks for EEG decoding and visualization: Convolutional Neural Networks in EEG Analysis,” Hum. Brain Mapp., vol. 38, no. 11, pp. 5391-5420, Nov. 2017.

---

> ### Author Response · Authors · 2024-11-21
> **Response [4/4] to Official Review by Reviewer ocKU - Questions [2/2]**
>
> > **D and P are both used for the data dimension**
>
> Actually, P refers to the channel dimension of the observed EEG data, while D refers to the dimensionality of latent SPD features that carry information about class labels.
> The feature extractor $f_{\theta}$ transforms P-channel EEG segments to points on the D-dimension SPD manifold.
> To improve clarity, we re-designed the overview figure (Figure 1) and rewrote section 3 of the revised manuscript.
>
> > **There are “?” in lines 218 and 236. Q and U are both used for the domain-invariant part of the mixing matrix.**
>
> Thank you for reporting these errors back to us. We fixed them in the revised manuscript along with other erros that we identified in the meantime.

---

> ### Comment · Reviewer_ocKU · 2024-11-25
>
> I sincerely thank the authors for addressing my concerns and providing detailed responses. The paper presents interesting ideas, some of which are novel, and offers fresh insights into domain adaptation for EEG applications. The numerical experiments provide valuable evidence of the method’s effectiveness, though the observed improvements are relatively modest. While the experiments demonstrate the potential of the approach, further work may be needed to fully establish its impact.
>
> Overall, I believe the paper makes a meaningful contribution to the field, and in my opinion, it deserves to be accepted. Considering the improvements made and the clarifications provided, I have increased my rating from 5 to 6.

---

> ### Author Response · Authors · 2024-11-28
> **New version updated within openreview due to discussion period extension**
>
> We extend our sincere gratitude for not only appreciating this work and enhancing the rating but also for your valuable insights and suggestions.
>
> As the discussion period has been extended for 6 days, we decided to include more experiments and update the latest manuscript within openreview.
> We would be grateful and honored to hear if there are still any potential improvements or minor concerns we can address to improve our rating.
>
> We mainly made the following changes in our latest submission within openreview.
>
> We added the extension simulation experiment results of varying predefined parameters in Appendix A.6. These results greatly enhance a deeper understanding of our proposed framework.
>
> For motor imagery experiments, we included more classical EEG models, including EEGNet (Lawhern+2018, *J. Neural Eng.* ), EEG-Conformer (Song+2022, *IEEE TNSRE*), ATCNet (Altaheri+2022, *IEEE Trans. Ind.Inform.*), and EEGInceptionMI (Zhang+2021, *J. Neural Eng.*).
> For sleep staing, we included DeepSleepNet (Supratak+2017, *IEEE TNSRE*) and AttnNet (Eldele+2021, *IEEE TNSRE*) as baseline methods.
> We additionally combine the competitive models with the multi-source SFUDA methods EA (He&Wu2019,*IEEE TBME*) and STMA (Gnassounou+2024,*arXiv*) within both settings.
> The additional motor imagery results are summarized in Table A4 (w/ label shifts) and Table A5 (w/o label shifts), and the additional sleep staging results are summarized in Table 1 and Table A2 in the latest manuscript revision.
>
> Altogether, the additional results clearly highlight the following:
> - TSMNet is a highly competitive architecture for motor imagery (especially cross-session transfer).
> - Although TSMNet was not initially proposed for sleep staging, the basic architecture is competitive with the highly specialized baseline deep learning architectures.
> - Our proposed SPDIM further boosts performance in the presence of label shifts.
>
> Since the empirical EEG data results for SPDIM are highly competitive - even after substantially increasing the considered baseline methods in motor imagery and sleep staging - we now have even stronger evidence that our modeling assumptions are suitable for real EEG data.
> Our experimental results clearly support our theoretical findings, indicating that geometric deep learning models like TSMNet have broader applicability.
>
> We are truly honored and delighted to see the significant enhancement of our work based on your thoughtful feedback, particularly regarding the sleep staging experiment. We are eager to hear more from you.
>
> *References*:
> - Lawhern, Vernon J., et al. "EEGNet: a compact convolutional neural network for EEG-based brain-computer interfaces." Journal of neural engineering 15.5 (2018): 056013.
>
> - Song, Yonghao, et al. "EEG conformer: Convolutional transformer for EEG decoding and visualization." IEEE Transactions on Neural Systems and Rehabilitation Engineering 31 (2022): 710-719.
>
> - Altaheri, Hamdi, Ghulam Muhammad, and Mansour Alsulaiman. "Physics-informed attention temporal convolutional network for EEG-based motor imagery classification." IEEE transactions on industrial informatics 19.2 (2022): 2249-2258.
>
> - Zhang, Ce, Young-Keun Kim, and Azim Eskandarian. "EEG-inception: an accurate and robust end-to-end neural network for EEG-based motor imagery classification." Journal of Neural Engineering 18.4 (2021): 046014.
>
> - Eldele, Emadeldeen, et al. "An attention-based deep learning approach for sleep stage classification with single-channel EEG." IEEE Transactions on Neural Systems and Rehabilitation Engineering 29 (2021): 809-818.
>
> - Supratak, Akara, et al. "DeepSleepNet: A model for automatic sleep stage scoring based on raw single-channel EEG." IEEE transactions on neural systems and rehabilitation engineering 25.11 (2017): 1998-2008.
>
> - H. He and D. Wu, “Transfer Learning for Brain-Computer Interfaces: A Euclidean Space Data Alignment Approach,” IEEE Trans. Biomed. Eng., vol. 67, no. 2, pp. 399-410, Feb. 2020, doi: 10.1109/TBME.2019.2913914.
>
> - T. Gnassounou, A. Collas, R. Flamary, K. Lounici, and A. Gramfort, “Multi-Source and Test-Time Domain Adaptation on Multivariate Signals using Spatio-Temporal Monge Alignment,” Jul. 19, 2024, arXiv: 2407.14303. url: http://arxiv.org/abs/2407.14303

---

### Official Review · Reviewer_ASJZ · 2024-11-02

**Soundness:** 3
**Presentation:** 1
**Contribution:** 3
**Rating:** 5
**Confidence:** 2

**Summary:**

The "SPDIM" paper introduces a novel geometric deep learning framework aimed at enhancing source-free unsupervised domain adaptation (SFUDA) for EEG data under both conditional and label shifts. By leveraging the symmetric positive definite (SPD) manifold and employing a parameter-efficient manifold optimization strategy, the proposed method, SPDIM, addresses significant generalization challenges in EEG data processing, especially where traditional Riemannian geometry methods fall short due to label shifts. SPDIM shows promising improvements across multiple EEG-based applications, including brain-computer interface tasks and sleep staging, demonstrating its efficacy over prior alignment frameworks

**Strengths:**

1.	The introduction of an SPD-manifold-constrained bias parameter is an advancement for tackling SFUDA in EEG.
2.	The framework has been applied effectively across different tasks, showcasing broad applicability.
3.	SPDIM outperforms conventional methods, showing its resilience under varying label distributions.

**Weaknesses:**

1.	The motivation behind addressing label shifts and domain gaps with SPDIM is somewhat implicit, without clearly laying out why these challenges necessitate the proposed framework.
2.	The paper contains an extensive number of equations and mathematical formulations in the main text, which can make the methodology difficult to follow.
3.	Although the paper compares SPDIM with several baselines, a broader set of comparisons, especially with newer unsupervised or semi-supervised EEG methods, could provide further insights into SPDIM’s performance and robustness.
4.	While SPDIM improves accuracy under domain shifts, the model’s interpretability remains limited.

I will reconsider my assessment after checking rebuttual.

**Questions:**

Plz go and check weaknesses

---

> ### Author Response · Authors · 2024-11-21
> **Response [1/1] to Official Review by Reviewer ASJZ**
>
> Thank you for taking the time to evaluate our submission and providing valuable feedback.
> Please find our detailed responses below.
>
> ***Weakness***
>
>
> > **The motivation behind addressing label shifts and domain gaps with SPDIM is somewhat implicit, without clearly laying out why these challenges necessitate the proposed framework.**
>
> Thank you for expressing your concern regarding presentation issues. We agree that the presentation in the submitted manuscript can be greatly improved.
> Based on the feedback of this and other reviewers, we rewrote substantial parts of sections 3 and 4 and created a new overview Figure.
> Although we decided to keep the overall motivation and section order, we hope that our modifications drastically improve clarity and resolve your concern.
>
> For your convenience, we decided to include copies of the revised manuscript [manuscript_revised.pdf](https://anonymous.4open.science/r/SPDIM-ICLR2025--B213/manuscript_revised.pdf) and another file [manuscript_diff_submitted_revised.pdf](https://anonymous.4open.science/r/SPDIM-ICLR2025--B213/manuscript_diff_submitted_revised.pdf) that highlights the changes compared to the sumitted manuscript in the anonymous repository.
>
> > **The paper contains an extensive number of equations and mathematical formulations in the main text, which can make the methodology difficult to follow.2**
>
> Thank you for your feedback.
> We have received mixed feedback from the reviewers. After careful considerations, we decided to keep the theoretical analysis in the main text. Still, we hope that our efforts to improve clarity in the revised manuscript ease your concern.
>
> > **Although the paper compares SPDIM with several baselines, a broader set of comparisons, especially with newer unsupervised or semi-supervised EEG methods, could provide further insights into SPDIM's performance and robustness.**
>
> Indeed, the feedback from all reviewers indicated that we should compare our framework with more baseline methods. Following this request, we decided to include additional multi-source SFUDA methods, including, EA (He&Wu2019,*IEEE TBME*) and STMA (Gnassounou+2024,*arXiv*). We additionally changed the wording to better delineate the difference between the previously proposed SPDDSBN (Kobler+2022,*NeurIPS*) method and our proposed SPDIM framework.
> To keep the comparison focused, we decided to exclude semi-supervised as well as single source/target domain UDA methods.
>
> *References*:
>
> - H. He and D. Wu, “Transfer Learning for Brain-Computer Interfaces: A Euclidean Space Data Alignment Approach,” IEEE Trans. Biomed. Eng., vol. 67, no. 2, pp. 399-410, Feb. 2020, doi: 10.1109/TBME.2019.2913914.
>
> - T. Gnassounou, A. Collas, R. Flamary, K. Lounici, and A. Gramfort, “Multi-Source and Test-Time Domain Adaptation on Multivariate Signals using Spatio-Temporal Monge Alignment,” Jul. 19, 2024, arXiv: 2407.14303. url: http://arxiv.org/abs/2407.14303
>
> - R. Kobler, J. Hirayama, Q. Zhao, and M. Kawanabe, “SPD domain-specific batch normalization to crack interpretable unsupervised domain adaptation in EEG,” in Advances in Neural Information Processing Systems, S. Koyejo, S. Mohamed, A. Agarwal, D. Belgrave, K. Cho, and A. Oh, Eds., Curran Associates, Inc., 2022, pp. 6219-6235. url: https://proceedings.neurips.cc/paper_files/paper/2022/file/28ef7ee7cd3e03093acc39e1272411b7-Paper-Conference.pdf
>
>
> > **While SPDIM improves accuracy under domain shifts, the model's interpretability remains limited.**
>
> Thank you for sharing your concern.
> To keep the manuscript focused, we decided to present our decoding framework along with theoretical considerations and several experiments with empirical data.
> The empirical success of our decoding framework indicates that our generative model seems appropriate for EEG data. At the time of submission, we decided to leave the explainability analysis for future work.
> Note, that an XAI framework (Kobler+2021, *IEEE EMBC*) can be utilized to transform the fitted model parameters $\Theta = \lbrace \theta, \phi, \psi \rbrace$ of TSMNet (Kobler+2022, *NeurIPS*) to interpretable spectral and spatial patterns.
>
> *References*:
>
> - R. Kobler, J. Hirayama, Q. Zhao, and M. Kawanabe, “SPD domain-specific batch normalization to crack interpretable unsupervised domain adaptation in EEG,” in Advances in Neural Information Processing Systems, S. Koyejo, S. Mohamed, A. Agarwal, D. Belgrave, K. Cho, and A. Oh, Eds., Curran Associates, Inc., 2022, pp. 6219-6235. url: https://proceedings.neurips.cc/paper_files/paper/2022/file/28ef7ee7cd3e03093acc39e1272411b7-Paper-Conference.pdf
>
> - R. J. Kobler, J.-I. Hirayama, L. Hehenberger, C. Lopes-Dias, G. Müller-Putz, and M. Kawanabe, “On the interpretation of linear Riemannian tangent space model parameters in M/EEG,” in Proceedings of the 43rd Annual International Conference of the IEEE Engineering in Medicine and Biology Society (EMBC), IEEE, 2021. doi: 10.1109/EMBC46164.2021.9630144.

---

> > ### Comment · Reviewer_ASJZ · 2024-11-26
> >
> > Thank you to the authors for their responses. I have carefully read all the content and have decided to maintain my current score.

---

> ### Author Response · Authors · 2024-11-28
> **New version updated within openreview due to discussion period extension**
>
> Thank you very much for your feedback. Your suggestions have significantly improved the quality of our submissions.
>
> We are deeply sorry that our revision did not meet your high standards,
> given that the discussion period has been extended by six days, we have decided to incorporate additional experiments and update the latest manuscript. If there are still any potential improvements or minor issues we could address to enhance our rating, we would be most grateful and honored to do so.
>
> We mainly made the following changes in our latest manuscript within openreview.
>
> For motor imagery experiments, we included more classical EEG models, including EEGNet (Lawhern+2018, *J. Neural Eng.* ), EEG-Conformer (Song+2022, *IEEE TNSRE*), ATCNet (Altaheri+2022, *IEEE Trans. Ind.Inform.*), and EEGInceptionMI (Zhang+2021, *J. Neural Eng.*).
> For sleep staing, we included DeepSleepNet (Supratak+2017, *IEEE TNSRE*) and AttnNet (Eldele+2021, *IEEE TNSRE*) as baseline methods.
> We additionally combine the competitive models with the multi-source SFUDA methods EA (He&Wu2019,*IEEE TBME*) and STMA (Gnassounou+2024,*arXiv*) within both settings.
> The additional motor imagery results are summarized in Table A4 (w/ label shifts) and Table A5 (w/o label shifts), and the additional sleep staging results are summarized in Table 1 and Table A2.
>
> Altogether, the additional results clearly highlight the following:
>
> - TSMNet is a highly competitive architecture for motor imagery (especially cross-session transfer).
> - Although TSMNet was not initially proposed for sleep staging, the basic architecture is competitive with the highly specialized baseline deep learning architectures.
> - Our proposed SPDIM further boosts performance in the presence of label shifts.
>
> Since the empirical EEG data results for SPDIM are highly competitive - even after substantially increasing the considered baseline methods in motor imagery and sleep staging - we now have even stronger evidence that our modeling assumptions are suitable for real EEG data.
> Our experimental results clearly support our theoretical findings, indicating that geometric deep learning models like TSMNet have broader applicability.
>
> We would like to thank you again for your previous feedback, particularly regarding clarifications, which significantly enhanced the quality of our submission. We look forward to hearing more from you.
>
> *References*:
> - Lawhern, Vernon J., et al. "EEGNet: a compact convolutional neural network for EEG-based brain-computer interfaces." Journal of neural engineering 15.5 (2018): 056013.
>
> - Song, Yonghao, et al. "EEG conformer: Convolutional transformer for EEG decoding and visualization." IEEE Transactions on Neural Systems and Rehabilitation Engineering 31 (2022): 710-719.
>
> - Altaheri, Hamdi, Ghulam Muhammad, and Mansour Alsulaiman. "Physics-informed attention temporal convolutional network for EEG-based motor imagery classification." IEEE transactions on industrial informatics 19.2 (2022): 2249-2258.
>
> - Zhang, Ce, Young-Keun Kim, and Azim Eskandarian. "EEG-inception: an accurate and robust end-to-end neural network for EEG-based motor imagery classification." Journal of Neural Engineering 18.4 (2021): 046014.
>
> - Eldele, Emadeldeen, et al. "An attention-based deep learning approach for sleep stage classification with single-channel EEG." IEEE Transactions on Neural Systems and Rehabilitation Engineering 29 (2021): 809-818.
>
> - Supratak, Akara, et al. "DeepSleepNet: A model for automatic sleep stage scoring based on raw single-channel EEG." IEEE transactions on neural systems and rehabilitation engineering 25.11 (2017): 1998-2008.
>
> - H. He and D. Wu, “Transfer Learning for Brain-Computer Interfaces: A Euclidean Space Data Alignment Approach,” IEEE Trans. Biomed. Eng., vol. 67, no. 2, pp. 399-410, Feb. 2020, doi: 10.1109/TBME.2019.2913914.
>
> - T. Gnassounou, A. Collas, R. Flamary, K. Lounici, and A. Gramfort, “Multi-Source and Test-Time Domain Adaptation on Multivariate Signals using Spatio-Temporal Monge Alignment,” Jul. 19, 2024, arXiv: 2407.14303. url: http://arxiv.org/abs/2407.14303

---

### Official Review · Reviewer_CvWN · 2024-11-04

**Soundness:** 3
**Presentation:** 2
**Contribution:** 2
**Rating:** 6
**Confidence:** 3

**Summary:**

The study addresses a source-free unsupervised domain adaptation problem and proposes SPDIM, a framework based on the SPD manifold.
SPDIM compensates for label shifts using proposed generative models, which prior Riemannian statistical alignment methods do not effectively handle.
Additionally, SPDIM applies the information maximization principle to learn domain-specific parameters.
Simulation experiments demonstrate its superiority under various levels of label shift, and empirical analysis on real EEG datasets shows that it outperforms previous approaches.

**Strengths:**

- The motivation is clear and easy to follow. SPDIM aims to address adaptation under label shifts, a common challenge in real-world EEG datasets.
  Theoretical analysis further explains the causes of deviations under label shifts.

- Simulation experiments qualitatively validate the benefits of SPDIM in the presence of label shifts.
   Cross-subject and cross-session experiments on motor and sleep-staging EEG datasets illustrate its superiority over existing alignment methods based on the SPD manifold.

**Weaknesses:**

- Some notations in equations seem confusing. For example, the index $j$ under $\sum$ may need to be $i$ in Eq. (2). The invertible mapping $upper$ is defined on $S$, but $upper^{-1}$ appears in Eq.(10).
Additionally, $j_i$ and $j$ use the same letter but with different meanings, which could lead to ambiguity. The notation $Q$ in Eq.(15) seems to appear without prior introduction.

- Some aspects of the method require further clarification. As mentioned in Line 249, the right-hand side of Eq. (15) is claimed to contain only domain-invariant terms. However, from my perspective, $C_i$ depends on the domain-specific
matrix $A_{j}$, as suggested by Eq. (13). According to Proposition 1, $ \bar{C} _ {j(i)} $ converges to $I_P$. These indicate that $Q$ is linked to $A_{j}$, which may not be domain-invariant. Additionally, the relationship between the information maximization approach introduced in Section 3.3 and SPDIM (bias) / SPDIM (geodesic) is unclear.

- To better demonstrate SPDIM’s effectiveness, it would be beneficial to compare it with additional statistical alignment methods beyond those based on the SPD manifold. This would provide a more comprehensive evaluation against existing approaches.

**Questions:**

- Is the domain-specific formard model $A_{j}$ learned from features of a specific domain, or is it predefined?
- How are domain specific parameters $\Phi_{j(i)}$ and the geodesic step-size parameters $\varphi_{j(i)}$ learned according to the proposed information maximization principle described in Section 3.3?
- Is there any relationship between the adaptation performance and predefined hyperparameters, such as the rank of $A$ and the number of domains within $\mathcal{D}_{s}$?

---

> ### Author Response · Authors · 2024-11-21
> **Response [1/2] to Official Review by Reviewer CvWN - Weaknesses**
>
> Thank you very much for your time to assess our submission and the provided feedback.
> Please find our detailed responses to the weaknesses and questions below.
>
> For your convenience, we decided to include copies of the revised manuscript [manuscript_revised.pdf](https://anonymous.4open.science/r/SPDIM-ICLR2025--B213/manuscript_revised.pdf) and another file [manuscript_diff_submitted_revised.pdf](https://anonymous.4open.science/r/SPDIM-ICLR2025--B213/manuscript_diff_submitted_revised.pdf) that highlights the changes compared to the sumitted manuscript in the anonymous repository.
>
> ***Weaknesses***
>
> >  - For example, the index $j$ under $\sum$ may need to be $i$ in Eq. (2).
>
> Thank you for reporting these errors back to us.
>
> >  - The invertible mapping $\mathrm{upper}$ is defined on $S$, but $\mathrm{upper}^{-1}$ appears in Eq.(10).
>
> Thanks for pointing us to this. We adjusted the corresponding text section.
>
> >  - Additionally, $j_i $and $j$ use the same letter but with different meanings, which could lead to ambiguity.
>
> We agree that the notation $j$ and $j_i$ could lead to ambiguity. Unfortunately, we could not come up with a better compact notation so far.
> As defined in section 2.1, $j$ is defined as the domain, and $j_i$ indicates the associated domain for observation $i$.
>
> >  - The notation $Q$ in Eq.(15) seems to appear without prior introduction.
>
> Before submission we streamlined notation with the aim to minimize confusion across symbols but missed to update some occurrences.
> For example, the symbols $Q$ and $U$ refer to the same variable (i.e., $Q = U$ ) in the submitted manuscript.
>
> We fixed these issues and others in the revised manuscript, and apologize for overseeing errors and spelling mistakes in the submitted manuscript.
>
>
> > **As mentioned in Line 249, the right-hand side of Eq. (15) is claimed to contain only domain-invariant terms. However, from my perspective, $C_i$ depends on the domain-specific matrix $A_j$, as suggested by Eq. (13). According to Proposition 1, $\bar{C}_{j(i)}$ converges to $I_P$. These indicate that $Q$ is linked to $A_j$, which may not be domain-invariant.**
>
> We think that his concern is caused because of our notation error (i.e., $Q = U$) that we introduced in the submitted manuscript (see also our response to your previous comment).
> We are deeply sorry for introducing this misleading error.
> We assume the domain-specific matrix $A_j$ consists of a rotation part $Q$ shared across domains and a domain-specific scaling part $\mathrm{exp}(P_j)$ in our model.
> After fixing this notation error in the revised manuscript, it is clear that the right hand side in eq. (15) only contains domain-invariant terms if there are no label shifts.
>
> > **Additionally, the relationship between the information maximization approach introduced in Section 3.3 and SPDIM (bias) / SPDIM (geodesic) is unclear.**
>
> We apologize for failing to fully articulate some conceptual ideas behind our approach in the submitted manuscript.
> Depending on the choice of the bias parameter to be optimized, we distinguish between SPDIM(bias) defined in (19) and SPDIM(geodesic) defined in equation (20).
> SPDIM(geodesic) can be considered a restricted version of SPDIM(bias) because its solution space is constrained to a geodesic instead of the entire SPD manifold.
>
> We rewrote large parts in sections 3 and 4 and created a new overview Figure to improve the presentation of our proposed framework. We hope that these changes resolve this concern.
>
>
> > **To better demonstrate SPDIM's effectiveness, it would be beneficial to compare it with additional statistical alignment methods beyond those based on the SPD manifold. This would provide a more comprehensive evaluation against existing approaches.**
>
> Indeed, the feedback from all reviewers indicated that we should compare our framework with more baseline methods. Following this request, we decided to include additional established multi-source SFUDA methods, including, EA (He&Wu2019,*IEEE TBME*) and STMA (Gnassounou+2024,*arXiv*). We additionally changed the wording to better delineate the difference between the previously proposed SPDDSBN (Kobler+2022,*NeurIPS*) method and our proposed SPDIM framework.
>
> *References*
>
> - H. He and D. Wu, “Transfer Learning for Brain-Computer Interfaces: A Euclidean Space Data Alignment Approach,” IEEE Trans. Biomed. Eng., vol. 67, no. 2, pp. 399-410, Feb. 2020, doi: 10.1109/TBME.2019.2913914.
>
> - T. Gnassounou, A. Collas, R. Flamary, K. Lounici, and A. Gramfort, “Multi-Source and Test-Time Domain Adaptation on Multivariate Signals using Spatio-Temporal Monge Alignment,” Jul. 19, 2024, arXiv: 2407.14303.
>
> - R. Kobler, J. Hirayama, Q. Zhao, and M. Kawanabe, “SPD domain-specific batch normalization to crack interpretable unsupervised domain adaptation in EEG,” in Advances in Neural Information Processing Systems, S. Koyejo, S. Mohamed, A. Agarwal, D. Belgrave, K. Cho, and A. Oh, Eds., Curran Associates, Inc., 2022, pp. 6219-6235.

---

> > ### Author Response · Authors · 2024-11-21
> > **Response [2/2] to Official Review by Reviewer CvWN - Questions**
> >
> > ***Questions***
> >
> > > **Is the domain-specific forward model $A_j$ learned from features of a specific domain, or is it predefined?**
> >
> > In our generative model, we assume that the domain-specific forward model $A_j$ is predefined but not observed.
> > We further assume that it consists of a rotation part $Q$ that is shared across domains and a domain-specific scaling part $\mathrm{exp}(P_j)$.
> >
> > Although our decoding framework does not explicitly estimate $A_j := Q \mathrm{exp}(P_j)$, under the scenario without label shift, our theoretical analysis (specifically, proposition 2) demonstrates that the alignment function $\tilde{m}_\phi$ can compensate the effect of $\mathrm{exp}(P_j)$.
> > Empirical results presented in prior work and in our submission, demonstrate the effectiveness of this decoding framework for EEG data.
> > Under additional label shifts, our theoretically motivated SPDIM framework yields significant empirical performance gains.
> >
> >
> > > **How are domain specific parameters $\Psi_{j(i)}$ and the geodesic step-size parameters $\varphi_{j(i)}$ learned according to the proposed information maximization principle described in Section 3.3?**
> >
> > We apologize for not fully conveying the implementation details of our approach in the submitted manuscript.
> > We use the entire target domain data to estimate the IM loss for the target domain, and optimize the bias parameter through gradient descent.
> > In the revised manuscript, we improved clarity by emphasizing source domain training and target domain adaptation in section 4, and creating a new overview Figure (Figure 1).
> >
> >
> > > **Is there any relationship between the adaptation performance and predefined hyperparameters, such as the rank of $A$ and the number of domains within $\mathcal{D}_s$?**
> >
> > Thank you for raising this question. We noted your request and will run additional simulations that investigate potential effects associated with the number of sources/channels $P$, the dimensionality $D$ of the subspace that encodes label infomration, and the number of domains $|\mathcal{J}_s|$.
> > We apologize for prioritizing other responses over this request; we intend to provide a detailed response within the next days.

---

> > > ### Author Response · Authors · 2024-11-25
> > > **Additional Response to Official Review by Reviewer CvWN - Questions**
> > >
> > > ## Response to Additional Simulations
> > >
> > > We are deeply sorry to keep you waiting.
> > >
> > > To investigate the relationship between the adaptation performance and some predefined hyper-parameters, we conducted additional simulations under a fixed class separability (i.e., `class_separability=3`). Following your request, we varied the following predefined hyper-parameters:
> > >
> > > [Rebuttal Figure 1](https://anonymous.4open.science/r/SPDIM-ICLR2025--B213/figures/fig_exp-sim_y-bacc_x-ratio_col-ninformative_kind-line.png): Additional simulations results. Same as Figure 2 in the manuscript but for a different number of informative dimensions encoding label information in $s$, as defined in (11). The `n_informative` parameter effectively defines the dimensionality of the label encoding subspace $D$
> > >
> > > [Rebuttal Figure 2](https://anonymous.4open.science/r/SPDIM-ICLR2025--B213/figures/fig_exp-sim_y-bacc_x-ratio_col-samplesperdomain_kind-line.png): Additional simulations results. Same as Figure 2 in the manuscript but for a different number of samples per domains $M_j \in \lbrace 100, 200, 400, 800\rbrace$.
> > >
> > > [Rebuttal Figure 3](https://anonymous.4open.science/r/SPDIM-ICLR2025--B213/figures/fig_exp-sim_y-bacc_x-ratio_col-n_domains_kind-line.png): Additional simulations results. Same as Figure 2 in the manuscript but for a different number of source domains $|\mathcal{J}_s| \in \lbrace 1, 3, 5, 7\rbrace$.
> > >
> > > Our proposed methods SPDIM(bias) [red line] and SPDIM(geodesic) [green line] generally outperform RCT [orange line] by maintaining a higher score for the same label ratio. We further highlight some important observations:
> > > -  **Rebuttal Figure 1**: As the number of informative source increases from 1 to 3, there is a slight improvement in performance within the source domain [blue line]. Concurrently, the overlap between SPDIM(bias) and SPDIM (geodesic) decreases, indicating that SPDIM(bias) becomes more effective with more informative sources.
> > > -  **Rebuttal Figure 2**: As the samples per domain increases, SPDIM(bias) beneftis most and is closely followed by SPDIM (geodesic). This trend agrees with increases in model complexity (i.e., SPIM(bias) fits an additional SPD matrix, while SPIM(geodesic) fits only an additional scalar parameter).
> > > -  **Rebuttal Figure 3**: Increasing the number of source domains does not affect RCT. While both SPDIM methods generally outperform RCT, wider confidence intervals (estimated with 100 repetitions) for lower label ratios indicate larger variability.

---

> > > > ### Comment · Reviewer_CvWN · 2024-11-26
> > > >
> > > > Thank you for the detailed responses and revisions. After reading the explanations and experiments, most of my concerns regarding why SPDIM performs well under label shifts have been addressed. As a result, I have decided to raise my score accordingly (5 to 6).

---

> > > > > ### Author Response · Authors · 2024-11-28
> > > > > **New version updated within openreview due to discussion period extension**
> > > > >
> > > > > We are deeply grateful for your kind recognition and the improved rating of our work. All your feedback, particularly regarding the simulation experiment, has been very helpful in refining our submission.
> > > > >
> > > > > As the discussion period is now extended by six days, we have decided to incorporate additional experiments and update the latest manuscript.
> > > > > We would appreciate further input on any potential improvements or minor concerns we can address to boost our rating.
> > > > >
> > > > > We mainly made the following changes in our latest manuscript within openreview.
> > > > >
> > > > > Thanks to your suggestion about varying predefined parameters, we added the extension simulation experiment results. These results greatly enhance a deeper understanding of our proposed framework.
> > > > >
> > > > > For motor imagery experiments, we included more classical EEG models, including EEGNet (Lawhern+2018, *J. Neural Eng.* ), EEG-Conformer (Song+2022, *IEEE TNSRE*), ATCNet (Altaheri+2022, *IEEE Trans. Ind.Inform.*), and EEGInceptionMI (Zhang+2021, *J. Neural Eng.*).
> > > > > For sleep staing, we included DeepSleepNet (Supratak+2017, *IEEE TNSRE*) and AttnNet (Eldele+2021, *IEEE TNSRE*) as baseline methods.
> > > > > We additionally combine the competitive models with the multi-source SFUDA methods EA (He&Wu2019,*IEEE TBME*) and STMA (Gnassounou+2024,*arXiv*) within both settings.
> > > > > The additional motor imagery results are summarized in Table A4 (w/ label shifts) and Table A5 (w/o label shifts), and the additional sleep staging results are summarized in Table 1 and Table A2.
> > > > >
> > > > > Altogether, the additional results clearly highlight the following:
> > > > >
> > > > > - TSMNet is a highly competitive architecture for motor imagery (especially cross-session transfer).
> > > > > - Although TSMNet was not initially proposed for sleep staging, the basic architecture is competitive with the highly specialized baseline deep learning architectures.
> > > > > - Our proposed SPDIM further boosts performance in the presence of label shifts.
> > > > >
> > > > > Since the empirical EEG data results for SPDIM are highly competitive - even after substantially increasing the considered baseline methods in motor imagery and sleep staging - we now have even stronger evidence that our modeling assumptions are suitable for real EEG data.
> > > > > Our experimental results clearly support our theoretical findings, indicating that geometric deep learning models like TSMNet have broader applicability.
> > > > >
> > > > > We are eager to hear more from you.
> > > > >
> > > > > *References*:
> > > > > - Lawhern, Vernon J., et al. "EEGNet: a compact convolutional neural network for EEG-based brain-computer interfaces." Journal of neural engineering 15.5 (2018): 056013.
> > > > >
> > > > > - Song, Yonghao, et al. "EEG conformer: Convolutional transformer for EEG decoding and visualization." IEEE Transactions on Neural Systems and Rehabilitation Engineering 31 (2022): 710-719.
> > > > >
> > > > > - Altaheri, Hamdi, Ghulam Muhammad, and Mansour Alsulaiman. "Physics-informed attention temporal convolutional network for EEG-based motor imagery classification." IEEE transactions on industrial informatics 19.2 (2022): 2249-2258.
> > > > >
> > > > > - Zhang, Ce, Young-Keun Kim, and Azim Eskandarian. "EEG-inception: an accurate and robust end-to-end neural network for EEG-based motor imagery classification." Journal of Neural Engineering 18.4 (2021): 046014.
> > > > >
> > > > > - Eldele, Emadeldeen, et al. "An attention-based deep learning approach for sleep stage classification with single-channel EEG." IEEE Transactions on Neural Systems and Rehabilitation Engineering 29 (2021): 809-818.
> > > > >
> > > > > - Supratak, Akara, et al. "DeepSleepNet: A model for automatic sleep stage scoring based on raw single-channel EEG." IEEE transactions on neural systems and rehabilitation engineering 25.11 (2017): 1998-2008.
> > > > >
> > > > > - H. He and D. Wu, “Transfer Learning for Brain-Computer Interfaces: A Euclidean Space Data Alignment Approach,” IEEE Trans. Biomed. Eng., vol. 67, no. 2, pp. 399-410, Feb. 2020, doi: 10.1109/TBME.2019.2913914.
> > > > >
> > > > > - T. Gnassounou, A. Collas, R. Flamary, K. Lounici, and A. Gramfort, “Multi-Source and Test-Time Domain Adaptation on Multivariate Signals using Spatio-Temporal Monge Alignment,” Jul. 19, 2024, arXiv: 2407.14303. url: http://arxiv.org/abs/2407.14303

---

### Official Review · Reviewer_PCcy · 2024-11-05

**Soundness:** 3
**Presentation:** 4
**Contribution:** 3
**Rating:** 8
**Confidence:** 3

**Summary:**

This study focuses on the realistic issue of label shifts in EEG across subjects and/or sessions (relative class proportions in target domains when source domains are class-balanced). Using theoretical analysis, it extends the SotA statistical alignment framework for handling distribution shifts in EEG to also include label shifts. The proposed SPDIM includes a domain-specific bias parameter estimated from unlabeled target data that reduces over-corrections done by the current SotA framework. Results on synthetic data and real-world EEGs demonstrate the value of SPDIM over baselines.

**Strengths:**

- Rigorous and clear presentation of technical details and full analytic workflow.
- This work is a great example of theory-guided methods design for EEG.
- Impactful choice of research problem - performance of EEG models under label shifts will remain a ubiquitous concern, both clinically and in the BCI space.

**Weaknesses:**

- (line 166) Q: Is the assumption of number of latent brain sources = number of observed scalp channels = P necessary or realistic?
- No discussion of study limitations and/or future directions.

**Questions:**

- Q: Does this framework treat one subject or one EEG recording as one source/target domain containing both/multiple class labels?
- Q: How does this framework for "latent space alignment" compare/relate to non-reimannian approaches for SFUDA for EEGs/multivariate timeseries? See [1] for a recent example. The "test-time adaptation" (Section 3.2) studies listed in [2] might also be relevant.
- Q: What factors other than dataset size and label shifts could account for the high variability/stdev in Table 1? In most cases, handling label shift (either with RCT or SPDIM) decreases variability compared to "w/o", but its still seems high.
- Minor comments: 1) pixel resolution of Figure 1 can be improved, 2) typo in citations at line 218 and 236., 3) line 443 remove "standard-deviation in brackets"
- The anonymous code link is broken?

[1] He, Huan, et al. "Domain adaptation for time series under feature and label shifts." International Conference on Machine Learning. PMLR, 2023.

[2] Garg, Saurabh, et al. "Rlsbench: Domain adaptation under relaxed label shift." International Conference on Machine Learning. PMLR, 2023.

---

> ### Author Response · Authors · 2024-11-21
> **Response [1/2] to Official Review by Reviewer PCcy  - weaknesses**
>
> Thank you very much for recognizing the merit in our submission and providing positive feedback along with your review.
> Please find our detailed responses to the weaknesses and questions below.
>
> ***Weakness***
>
> > **(line 166) Q: Is the assumption of number of latent brain sources = number of observed scalp channels =  P  necessary or realistic?**
>
> Thank you for your question.
> No, the assumption is not necessary.
> We are sorry for not presenting this model assumption clearly in our framework.
> Like prior work (for example: Sabbagh et al. 2020, *NeuroImage*), we actually assume that the latent sources whose covariance encodes label information are constrained to a submanifold $\mathcal{S}_D^+$ with $D \le P$, and $P$ representing the number of latent sources / EEG channels.
>
> Generally, data-driven models (e.g., independent component analysis) frequently assume that the forward model as an invertible linear transformation (i.e., the number of observed channels is similar to the number of latent brain sources).
>
> To emphasize this assumption, we rewrote section 3.1 and created a new overview Figure in the revised manuscript.
>
> *References:*
>
> Sabbagh, David, et al. "Predictive regression modeling with MEG/EEG: from source power to signals and cognitive states." NeuroImage 222 (2020): 116893.
>
> > **No discussion of study limitations and/or future directions.**
>
> We are sorry for not including these in the submitted manuscript.
> A limitation of our framework is that the IM loss, due to large noise and outliers, can sometimes estimate an inappropriate bias parameter, leading to the data being shifted in the wrong direction.
> We hope future work will explore more robust methods for estimating the bias parameter.
> We added a brief discussion about this limitation in the revised manuscript.

---

> ### Author Response · Authors · 2024-11-21
> **Response [2/2] to Official Review by Reviewer PCcy  - Questions**
>
> ***Questions***
>
> > **Does this framework treat one subject or one EEG recording as one source/target domain containing both/multiple class labels?**
>
> Thank you for your question.
> We consider the closed-set SFUDA problem, so the class labels are consistent in source and target domains.
> We treat one session as one source/target domain. In the cross-subject scenarios, all sessions of subjects in the test set are considered as target domains. In the cross-session scenario, we fit models per subject and split sessions into source and target domains.
>
>
> > **Q: How does this framework for "latent space alignment" compare/relate to non-riemannian approaches for SFUDA for EEGs/multivariate timeseries? See [1] for a recent example. The "test-time adaptation" (Section 3.2) studies listed in might also be relevant.**
>
> Indeed, the feedback from all reviewers indicated that we should compare our framework with more baseline methods. Following this request, we decided to include additional multi-source SFUDA methods, including, EA (He&Wu2019,*IEEE TBME*) and STMA (Gnassounou+2024,*arXiv*). We additionally changed the wording to better delineate the difference between the previously proposed SPDDSBN (Kobler+2022,*NeurIPS*) method and our proposed SPDIM framework.
> To keep the comparison focused, we decided to exclude semi-supervised as well as single source/target domain UDA methods (like RAINCOAT proposed in [1]).
>
> *References*:
>
> - H. He and D. Wu, “Transfer Learning for Brain-Computer Interfaces: A Euclidean Space Data Alignment Approach,” IEEE Trans. Biomed. Eng., vol. 67, no. 2, pp. 399-410, Feb. 2020, doi: 10.1109/TBME.2019.2913914.
>
> - T. Gnassounou, A. Collas, R. Flamary, K. Lounici, and A. Gramfort, “Multi-Source and Test-Time Domain Adaptation on Multivariate Signals using Spatio-Temporal Monge Alignment,” Jul. 19, 2024, arXiv: 2407.14303. url: http://arxiv.org/abs/2407.14303
>
> - R. Kobler, J. Hirayama, Q. Zhao, and M. Kawanabe, “SPD domain-specific batch normalization to crack interpretable unsupervised domain adaptation in EEG,” in Advances in Neural Information Processing Systems, S. Koyejo, S. Mohamed, A. Agarwal, D. Belgrave, K. Cho, and A. Oh, Eds., Curran Associates, Inc., 2022, pp. 6219-6235. url: https://proceedings.neurips.cc/paper_files/paper/2022/file/28ef7ee7cd3e03093acc39e1272411b7-Paper-Conference.pdf
>
> > **Q: What factors other than dataset size and label shifts could account for the high variability/stdev in Table 1? In most cases, handling label shift (either with RCT or SPDIM) decreases variability compared to "w/o", but its still seems high.**
>
> Interesting question.
> Typically model performance varies greatly among individual subjects, leading to high standard deviation across subjects.
> Because we wanted to test generalization across subjects, we calculated the summary statistics (mean and standard deviation) at the subject level. Specifically, we computed the balanced accuracy metric for each subject in the test set individually. In doing so, we obtained a balanced accuracy score per subject after cross-validation was completed. The list of scores was then used to compute the summary statistics. Additionally, we computed paired t-tests to compare the performance of methods while controlling for the variability across subjects.
>
> > **Minor comments**
> >  1. pixel resolution of Figure 1 can be improved,
> >  2. typo in citations at line 218 and 236.,
> >  3. line 443 remove "standard-deviation in brackets"
>
> Thank you for reporting these back to us.
> We apologize for introducing several typos and inconsistencies in the submitted manuscript.
> We fixed them in the revised manuscript along with other erros that we identified in the meantime.
>
> For your convenience, we decided to include copies of the revised manuscript [manuscript_revised.pdf](https://anonymous.4open.science/r/SPDIM-ICLR2025--B213/manuscript_revised.pdf) and another file [manuscript_diff_submitted_revised.pdf](https://anonymous.4open.science/r/SPDIM-ICLR2025--B213/manuscript_diff_submitted_revised.pdf) that highlights the changes compared to the sumitted manuscript in the anonymous repository.
> > **The anonymous code link is broken?**
>
> We apologize for the oversight that caused the broken code link at that time. We updated the code link in the revised manuscript: https://anonymous.4open.science/r/SPDIM-ICLR2025--B213

---

### Meta-Review · Area_Chair_Gy9D · 2024-12-16

**Metareview:**

This paper was considered "Rigorous" with "clear presentation", and "a great example of theory-guided methods design for EEG". by reviewer PCcy and also endorsed by reviewers CvWN and ocKU, in particular after a valuable discussion with ocKU that surely helped clarify the paper and increase it's experimental part.

Based on reviews and further discussions, this paper is considered a relevant and good contribution for the ICLR community and in particular the ML researchers working on neural signal decoding.

**Additional Comments On Reviewer Discussion:**

reviewer PCcy and also endorsed by reviewers CvWN and ocKU, in particular after a valuable discussion with ocKU that surely helped clarify the paper and increase it's experimental part.

---

### Decision · Program_Chairs · 2025-01-22

Accept (Poster)